# Large ice loss variability at Nioghalvfjerdsfjorden Glacier, Northeast-Greenland

Christoph Mayer [1], Janin Schaffer [2], Tore Hattermann [2,3], Dana Floricioiu [4], Lukas Krieger [4], Paul A. Dodd [5], Torsten Kanzow[2], Carlo Licciulli[1] & Clemens Schannwell[6]

Nioghalvfjerdsfjorden is a major outlet glacier in Northeast-Greenland. Although earlier studies showed that the floating part near the grounding line thinned by 30% between 1999 and 2014, the temporal ice loss evolution, its relation to external forcing and the implications for the grounded ice sheet remain largely unclear. By combining observations of surface features, ice thickness and bedrock data, we find that the ice shelf mass balance has been out of equilibrium since 2001, with large variations of the thinning rates on annual/multiannual time scales. Changes in ice flux and surface ablation are too small to produce this variability. An increased ocean heat flux is the most plausible cause of the observed thinning. For sustained environmental conditions, the ice shelf will lose large parts of its area within a few decades and ice modeling shows a significant, but locally restricted thinning upstream of the grounding line in response.

[1] Bavarian Academy of Sciences and Humanities, Alfons-Goppel Str. 11, 80539 Munich, Germany. [2] Alfred Wegener Institute, Helmholtz Centre for Polar and Marine Research, Am Handelshafen 12, 27570 Bremerhaven, Germany. [3] Akvaplan-niva AS, Fram Centre, Postbox 66069296 Langnes, Tromsø, Norway. [4] Remote Sensing Technology Institute, German Aerospace Centre (DLR), Münchener Straße 20, 82234 Oberpfaffenhofen, Weßling, Germany. [5] Norwegian Polar Institute, Fram Centre, Postbox 66069296 Langnes, Tromsø, Norway. [6] University of Tübingen, Geologie & Geodynamik, Wilhelmstraße 56, 72074 Tübingen, Germany. Correspondence and requests for materials should be addressed to C.M.(email: christoph.mayer@keg.badw.de)

Nioghalvfjerdsfjorden Glacier or 79 North Glacier has the largest ice shelf in Greenland with a length of more than 70 km and a width of about 20 km at mid-distance. Together with its neighbors Zachariæ Isstrøm and Storstrømmen, it is one of the major outlets of the North East Greenland Ice Stream (NEGIS), sharing a catchment area of almost 200,000 km² or 12% of the Greenland ice sheet area[1]. This region has the potential to rise global sea level by 1.1 m in the unlikely case of complete loss of this ice sheet sector[1]. The region also represents a major transport route for ice discharge into the Nordic Seas where it adds to the oceanic freshwater budget and large-scale circulation (for example see ref.[2]).

While other regions of the Greenland ice margin have already shown strong mass loss[3], the mass balance of the upper parts of NEGIS was long considered to be close to equilibrium (for example see ref.[4]), while thinning was observed closer to the margin. It was assumed that the region was not much influenced by climate change until the beginning of the new millennium[5,6]. The area of 79 North Glacier has remained remarkably stable since the first observations of its calving front in 1906[7], with no signs of ice shelf break-up even during recent years. However, the upstream NEGIS sector has shown increasing thinning rates since 2006[7] and Zachariæ Isstrøm has experienced the disintegration of its frontal ice shelf and an increase in ice flux of 50% between 1976 and 2015. Currently, Zachariæ Isstrøm loses about 5 Gt year⁻¹[1]. Even though the impact of changes in environmental parameters is not known in detail, the increased thinning rates are likely related to increased air temperatures leading to higher melt rates and a reduction in summer sea ice concentration. This facilitates higher calving and retreat rates, associated with a positive feedback due to a retrograde bed slope and reduction in buttressing from the glacier margins. In addition, the entry of warm subsurface ocean water could intensify the mass loss by melting[6].

This recent evolution of Zachariæ Isstrøm and its potential causes raises the imminent question about the future stability of its northern neighbor, the 79 North Glacier's floating part. Recently observed warming of Atlantic Waters in the Nordic Seas (e.g., in Fram Strait[8]) and the Arctic Ocean[9], increasing Arctic surface air temperatures[10] and more regular fast ice summer breakups since 2001[11] might affect this ice shelf in a similar way. The loss of the ice shelf might imply a reduction of the buttressing of the ice sheet, leading to enhanced ice discharge, progressive thinning and retreat of the grounding line in dependence of the bedrock geometry[12]. The ice sheet thinning at the grounding line leads to larger surface slopes and thus to enhanced ice flow, which will gradually spread further upstream until a new balance geometry is reached.

The floating part of 79 North Glacier fills the fjord between Kronprins Christian Land in the North and Lambert Land in the South (Fig. 1). Seismic measurements have revealed a deep ocean cavity beneath the ice shelf[13] that extends down to 900 m below sea level near the grounding line and rises eastwards towards the calving front, where a sill culminates in several shallow bedrock highs, which are pinning the ice shelf.

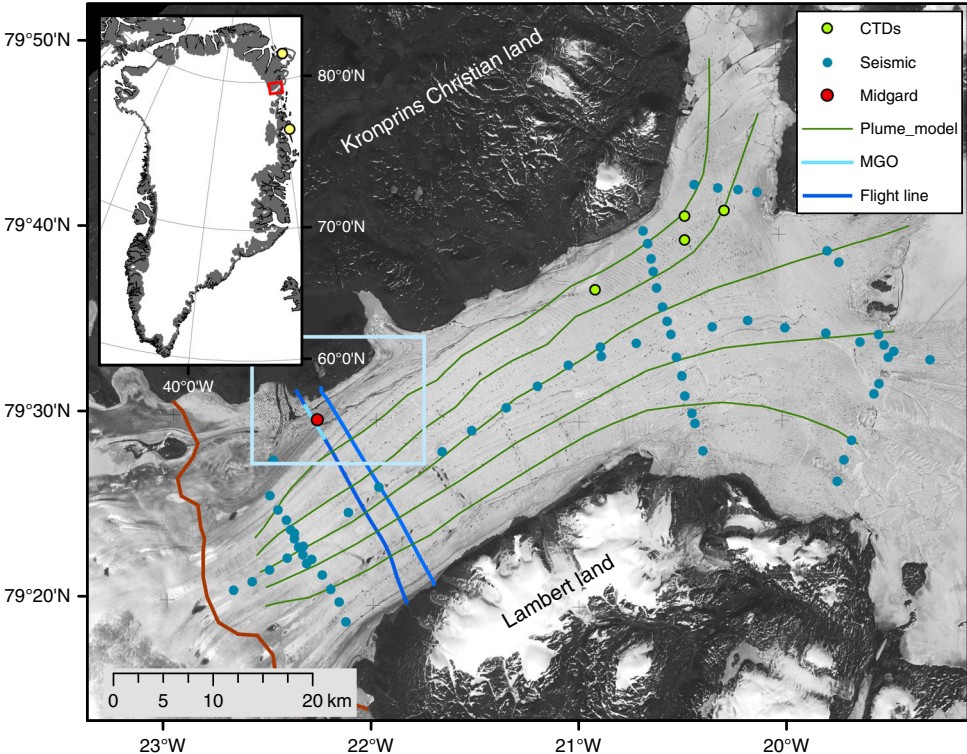

**Fig. 1** The floating part of 79 North Glacier with the data sets used in the study. 79 North Glacier in Northeast Greenland, with the locations of the cross profiles used in this analysis (dark blue: Radar and IceBridge flight lines, light blue: MGO cross profile as in Figs. 2,4 and 6). The location of the Midgardsormen experiment is shown as a red dot, while the seismic measurements from 1997 and 1998 are shown as blue dots[13]. The light green dots represent the CTD profiles used in this study. The green lines represent transects used for the plume model simulations. The light blue box shows the geographical extent of Fig. 4 and the upstream part of the MGO-ridge (gray winding feature). The grounding line is indicated by the red line in the lower left corner. The ice shelf front is located to the right of the image and towards the North (Djimphna Sund, at the upper right part of the image). The background image is taken from a Landsat 5 scene from 25 July 1998. The locations of the weather stations in Danmarkshavn and Station Nord are shown in the inset map as yellow dots

An ice ridge along the northern boundary of the ice shelf represents a remarkable surface feature of the glacier. It was named Midgardsormen (MGO-ridge) after the disappeared middle earth snake in Nordic mythology, due to its resemblance of a winding snake. The location of Midgardsormen and of the different data sets used in this study are shown in Fig. 1. The fact that MGO-ridge is not stable in time lead to the hypothesis that its migration is linked to changes in ice thickness and can be used to construct a time series of the ice shelf thickness evolution. We analyze the impact of possible forcing mechanisms to identify the main drivers of the observed thinning and use an ice dynamic model to investigate the response of the grounded ice. Oceanic energy transport is the most likely source for the strong variability of the observed ice thickness changes. We find that the floating part of the glacier very likely will disappear during the coming decades, while the effect on the adjacent grounded ice is significant, but locally restricted.

## Results

### The lateral grounding line of Midgardsormen.
To date, no detailed analysis of temporal ice thickness changes are available for 79 North Glacier, apart from short period remote sensing observations (for example see ref.[14]). Here, we present a time series of ice thickness changes dating back to 1998, with a temporal resolution of better than three years.

The migration of the MGO-ridge is utilized to investigate thickness changes of the floating ice tongue over time with an approximately annual temporal resolution (Table 1). The detailed geophysical measurements across Midgardsormen in 1998 reveal that it represents a special type of grounding line delineation of ref.[15]) re-grounding at a shallow angle with respect to ice flow on a lateral bedrock shoulder in the fjord (Fig. 2b). This results in a peculiar pressure ridge at the surface, which we appropriately name Midgardsormen (Fig. 2a). Results from the seismic measurements in Fig. 3a show that the bedrock rises from the central trough just south of Midgardsormen and then forms a gently sloping, shallow plain. A transition from floating to grounded ice north of Midgardsormen (to the left of 2380 m on the x-axis of Fig. 3) is confirmed by the dampening of the tidal tilt across the ice ridge (Supplementary Table 1). The amplitude of the tidal signal, recorded at the tiltmeter site of NF1 located at 383 m south of Midgardsormen on the floating ice shelf (Fig. 2b), is reduced by a factor of 10 at the center of the ridge and by a

factor of 30 at a distance of 185 m on the northern flank. The lateral strain, associated with the strong transversal ice velocity gradient is responsible for the formation of the MGO-pressure ridge that delineates the grounding line position. The exact width of the subglacial bedrock shoulder between the location of the Midgardsormen experiment and the seismic cross profile about 37 km further downstream is unclear, but the continuous surface expression of the ice ridge on the Landsat images (Fig. 4) suggests an extent of several kilometers to the East. The relatively smooth ice surface elevation of the freely floating central part of the glacier south of the MGO-ridge suggests that the ice shelf is in hydrostatic equilibrium, such that the depth of the ice draft below sea level is a direct measure of the local ice thickness. In the following, we utilize the fact that small changes in ice thickness lead to large movements of the grounding line over the gently sloping bedrock, to reconstruct the history of ice shelf thickness.

### Grounding line migration and related thickness changes.
Co-registered Landsat scenes reveal that the position of Midgardsormen moved about 2.1 km towards NW from 1994 to 2014 (Fig. 4). The displacements between the acquisition times of all Landsat scenes are given in Table 1, showing a successive northward (up-slope) displacement of the MGO-ridge, which is consistent with a continuous thinning of the ice shelf. The basal topography of the subglacial fjord shoulder north of the Midgardsormen position in 1998 is known from the seismic measurements near the ridge and the airborne radar ice thickness measurements towards the northern ice margin (Fig. 3). Based on this information, we estimate the observed northward migration of Midgardsormen between 1998 and 2014 to correspond to an ice thickness reduction of 85.9 m, or a mean ice thickness loss of 5.3 m year$^{-1}$ during this period.

The Landsat images show that Midgardsormen had moved another 682 m between 1994 and 1998. This indicates an earlier onset of the thinning, although no measurements are available for the fjord bottom at the location of the 1994 grounding line, which inhibits the quantification of ice thickness change in this early period.

In order to relate these observations to the larger region, we compare ice shelf thicknesses from ground penetrating airborne radar (cross profile 365000–392000, Fig. 1, western profile) and buoyancy derived ice thicknesses from TanDEM-X surface elevations along the same profile, south of the 1998 Midgardsormen position. Details about the used TanDEM-X scenes (Supplementary Table 2) and the resulting thickness changes (Supplementary Table 3) are provided in the Supplementary Information. Uncertainties connected to the TanDEM-X elevation calculations are presented in the Methods section. The results show a mean ice thickness reduction of 89.5 m between 1997 and 2014. Moreover, the ice shelf lost another 8.6 m according to surface elevation changes from TanDEM-X data that are analyzed over a larger floating area between December 2014 and September 2016. The comparison with surface elevation data from Operation IceBridge ATM data results in similar differences (Supplementary Table 4). The mean annual ice thickness loss was about 5.16 m year$^{-1}$ for the entire period. Additionally, surface elevation changes over the entire floating part of the ice shelf have been calculated from overlapping TanDEM-X acquisitions. The resulting ice thickness changes in the time periods 2011–2012, 2012–2014, and 2014–2016 are found in the Fig. 5 (numerical values in Supplementary Table 3).

The total ice thickness change, based on the cross profile comparison, is almost identical to the magnitude derived from the total lateral grounding line displacement. In the following, we thus assume that the local changes in ice thickness of higher

**Table 1 Midgardsormen grounding line migration**

| Date of landsat scene | Total displacement (m) | Ice thickness (m) | Relative thickness change (m/year) |
|---|---|---|---|
| 14-08-1994 | −682 ± 60 | | |
| 25-07-1998 | 0 | 282 ± 6.4 | |
| 26-06-2001 | 391 ± 60 | 282 ± 6.4 | −0.14 ± 2.2 |
| 10-06-2002 | 531 ± 60 | 269 ± 6.4 | −12.83 ± 6.7 |
| 20-07-2005 | 841 ± 60 | 243 ± 6.4 | −8.28 ± 2.1 |
| 31-07-2006 | 911 ± 60 | 242 ± 6.4 | −1.15 ± 6.2 |
| 11-08-2007 | 946 ± 60 | 240 ± 6.4 | −2.69 ± 6.2 |
| 08-08-2009 | 998 ± 60 | 236 ± 6.4 | −1.79 ± 3.2 |
| 16-07-2010 | 1076 ± 60 | 224 ± 6.4 | −13.10 ± 6.8 |
| 07-07-2012 | 1227 ± 60 | 209 ± 6.4 | −7.41 ± 3.2 |
| 12-07-2014 | 1402 ± 60 | 196 ± 6.4 | −6.50 ± 3.2 |

Observed northward displacement of Midgardsormen between 1994 and 2014 based on the available Landsat scenes. Ice thickness and its changes are derived from the displacement, the known bed topography and the floatation criterion of the ice shelf. The error calculations are discussed in the Methods section

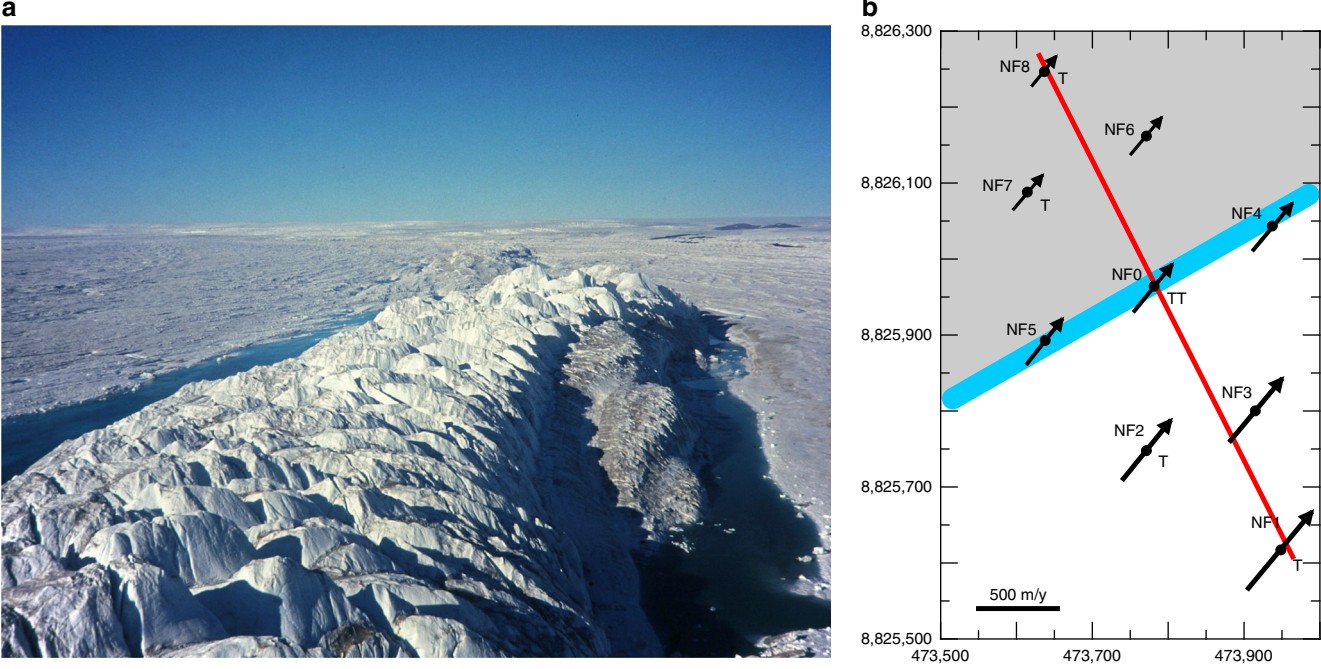

**Fig. 2** Migardsormen ice ridge and the local measurements. **a** Ice ridge of Midgardsormen on the northern part of the ice shelf, close to the location of the ice ridge measurements in the 1990s. The view is upstream towards the grounding line of 79 North Glacier (photo: C. Mayer, 1998). **b** measurements of the Midgardsormen experiment. The blue line represents the ice ridge, the red line the seismic profile. Ice velocities are displayed as black arrows and the positions of tilt meters (T) are indicated. The grounded part is indicated by gray shading

temporal resolution, inferred from the grounding line migration, are representative of the mean regional thickness changes along the entire floating part of the cross profiles. Across the entire floating part of the cross profile an ice thickness loss of 26% was detected from 1998 to 2014 (mean ice thickness in 1998: 338.2 ± 20.1 m), while the corresponding thickness change at the grounding line is 30%.

Based on the analysis of the grounding line migration, the thickness change reveals the following pattern: In the late 1990s, the floating ice tongue seems to be close to an equilibrium with a minor change in ice thickness (in spite of the observed considerable grounding line migration between 1998 and 2001). In the first years of the new millennium, the situation changes considerably and an ice thickness loss of more than 12 m is derived for 2001/2002. After this period, the thinning gradually reduces to rather low values of 1.5 m year$^{-1}$ during 2006–2009. Between 2009 and 2012 the annual ice thinning intensifies again, reaching 9.0 m year$^{-1}$ for the entire period (more than 12 m year$^{-1}$ in 2009/2010), before it reduces again slightly to 6.5 m year$^{-1}$ between 2012 and 2014. The analysis of the recent TanDEM-X elevation models provides additional information for the period 2014 until 2016, where the ice thickness is reduced by another 5.2 m year$^{-1}$ in the region of Midgardsormen.

**Potential reasons for the ice thickness variations**. There are several potential reasons for this considerable volume loss of the ice shelf. A change in the horizontal velocity structure (e.g., a slow down near the grounding line and/or an acceleration towards the calving front) could lead to dynamic thinning of the floating ice. Also, variations in the surface mass balance could induce higher thinning rates and thus a reduction of the ice thickness. Increasing melt rates, however, might also be induced by changes in the subglacial oceanic conditions (such as ocean warming).

Ice velocities showed almost no change during the entire period since 1998. The mean ice velocity along the ice shelf cross profile at Midgardsormen, determined with the IMCORR correlation algorithm[16], amounts to 859 ± 10 m year$^{-1}$ for the period 1998–2001 and to 843 ± 15 m year$^{-1}$ for 2009 until 2010. Also the along flow velocity gradients did not change significantly over this period, indicating that the calculated ice thickness change is not related to dynamic thinning of the glacier. The strain rate in the center of the ice shelf, south of Midgardsormen and between the two ice shelf cross profiles in Fig. 1 (blue lines), shows only a very small increase from −0.0357 ± 0.0263 year$^{-1}$ in 1998 to −0.0374 ± 0.0278 year$^{-1}$ in 2009. This relates to an increase in dynamic thickening of 0.56 ± 0.43 m year$^{-1}$ (from 11.78 ± 8.68 m year$^{-1}$ to 12.34 ± 9.18 m year$^{-1}$, respectively). Therefore, a change in ice dynamics cannot be the main reason for the observed ice thickness reduction of 91.1 m between 1998 and 2016.

To estimate the role of variations of the surface mass balance for the temporal variability of the ice thickness, we evaluated potential surface melt magnitudes based on temperature records from the closest weather stations Danmarkshavn and Station Nord. However, only Danmarkshavn weather station provides a continuous temperature record from 1958 until now[17]. The recorded summer air temperatures indicate a tendency to higher surface melt rates in the recent years (Supplementary Fig. 1). The surface melt rate (SMR) is based on the positive degree day sums (PDD) per year and computed by

$$\text{SMR} = k_{\text{ice}}(\text{PDD} - \text{PDD}_{0.5msnow}) + k_{\text{snow}}\text{PDD}_{0.5msnow}. \qquad (1)$$

The degree day factor for melting glacier ice ($k_{\text{ice}}$) is taken to be $k_{\text{ice}} = 9.6$ mm K$^{-1}$ based on measurements by[18] on Storstrømmen Glacier (Northeast Greenland). The degree day factor for melting snow ($k_{\text{snow}}$) is taken to be 40% of $k_{\text{ice}}$ in accordance to[19]. We assume an average snow cover of 0.5 m thickness that needs to be

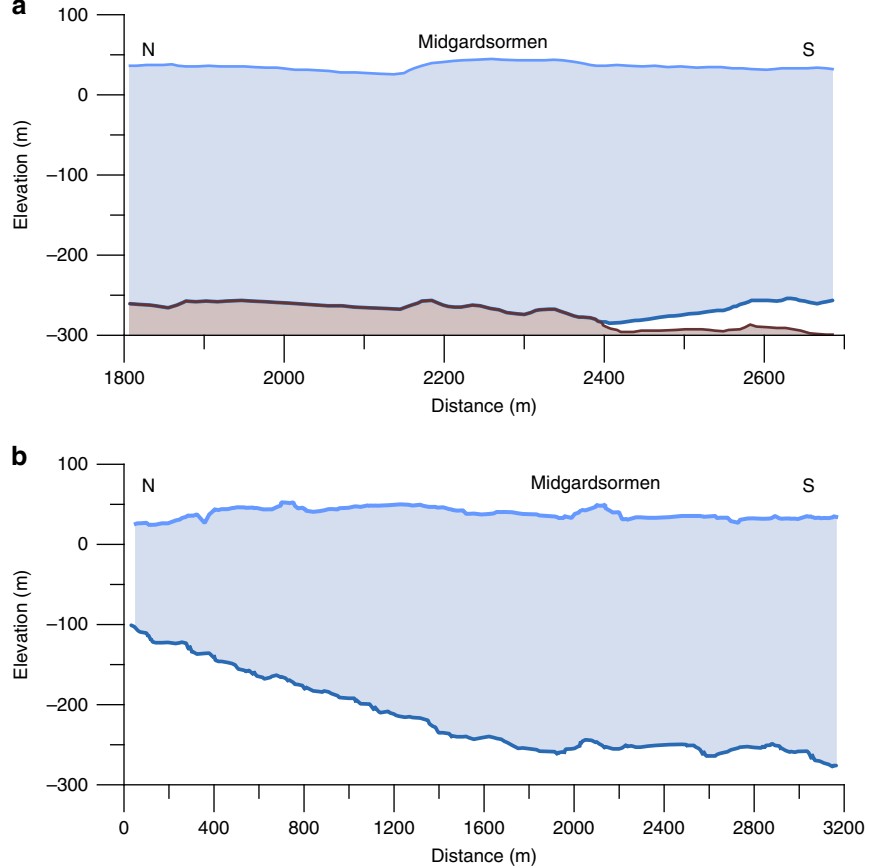

**Fig. 3** Glacier geometry along the cross section of GPR and seismic transects. **a** Cross section of Midgardsormen ridge, as revealed by the seismic measurements in 1998 (location: red dot in Fig. 1). **b** Glacier geometry along the first 3.2 km of the light blue profile in Fig. 4, according to the airborne radar measurements in 1997. The x-axis coordinates are identical in **a** and **b**. The y-axis in **a** represents the true scale, while the vertical axis is exaggerated three-fold in **b**

melted before glacier ice is melting. Based on these assumptions, we find that the maximum year-to-year variation of surface ice melt is 1.4 m year$^{-1}$ only (Supplementary Fig. 1). Even though there is a tendency to higher surface melt rates especially after 2000, the variations in surface melt rates are generally too small to explain the observed ice thickness changes derived from our analysis above. Also the high surface melt rate in 2008 is not reflected in the ice thickness evolution that was derived from the Midgardsormen migration.

Next, we consider the influence of changes in oceanic forcing on basal melting of the ice shelf. It has been shown[20] that the cavity is filled by a lower layer of Atlantic Water with maximum temperatures around +1 °C and an upper layer of colder and fresher Polar Water on top. Since the depth of the temperature/salinity gradient that separates these two layers coincides with the depth of the MGO-grounding line, i.e., ranging between 170 and 250 m (not shown), a thinning of the ice shelf would lead to a decrease in ocean temperature at the ice-ocean interface, if ocean properties remained unchanged. In contrast, the succession of CTD profiles shows an overall warming and thickening of the Atlantic Water layer in the cavity, such that temperatures at the respective depth of the MGO-grounding line increased by 0.2 °C between 1998 and 2014, despite its migration to shallower depth (Fig. 6). At 175 m depth, i.e., the depth at the grounding line position in 2014, temperatures increase from 0 to 0.5 °C. The evolution in the ice shelf cavity is consistent with hydrographic observations that show coherent warming of the Atlantic water

along Norske Trough - the main pathway across the continental shelf of Northeast Greenland from the shelf break in Fram Strait towards 79 North Glacier[21].

The effects of the observed ocean warming on the ice shelf basal mass loss are assessed with a simple, but well established ice-shelf plume model[22]. The area averaged, ensemble-mean basal melt rates from the model yield 8.7 ± 1.1 m year$^{-1}$ for 1998 and 12.2 ± 1.6 m year$^{-1}$ for 2014, which corresponds to a 40% increase in basal melt during that period. To translate the basal melt rate distribution along the plume path into an accumulative thinning of the ice shelf along a flowline, we calculate the path integral of basal mass loss for an ice column that is advected from the grounding line about 20 km downstream toward the approximate point where the glacier thinning was observed (i.e., at Midgardsormen (Fig. 1)). For that purpose, spatially varying ice flow velocities from the 2000/2001 MEaSUREs Greenland Ice Velocity Map[5,23] were interpolated onto each of the five ice base profiles used for the plume model (Supplementary Fig. 2), and integrations for each profile were repeated with eleven different starting points from the grounding line and downstream, shifted in 1-km steps. The advective time scale towards the Midgardsormen cross-flow profiles is approximately 20 years. Assuming that the effects of strain thinning and surface mass balance on the ice thickness evolution remain unchanged, the difference in accumulative thinning using either 1998 or 2014 melt rates gives the melt induced ice mass loss of the glacier for an instantaneous and sustained ocean warming. The results yield a

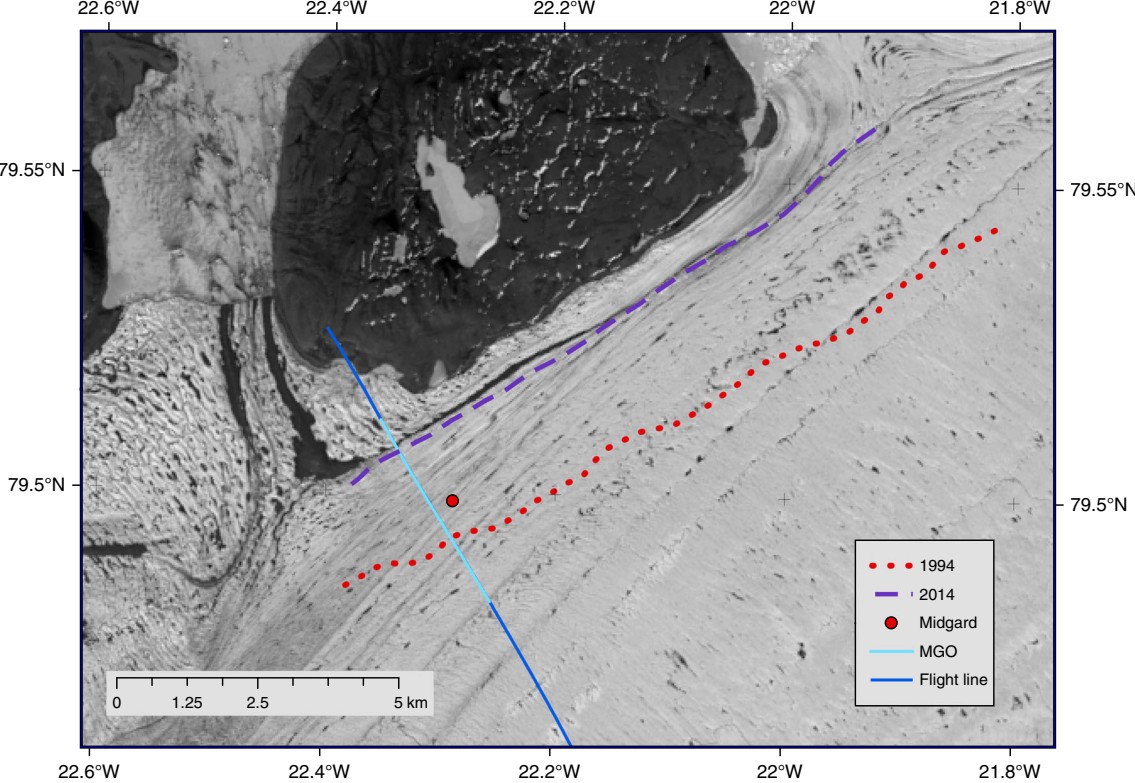

**Fig. 4** Migration of Midgardsormen from 1994 until 2014. Position of Midgardsormen in 1994 (red dotted line) and 2014 (dark purple dashed line). The red dot represents the location of the seismic experiment at Midgardsormen in 1998, while the blue line indicates the flight line of the airborne radar in 1997 and the elevation profile of the Icebridge ATM in 2012 (background image: Landsat 8 scene from 12 Jul 2014). The light blue section of the flight line represents the part displayed in Fig. 3b

total ensemble-mean ocean induced ice thickness loss of $61 \pm 20$ m. The estimate is comparable to the observed thinning of $86 \pm 6.4$ m at Midgardsormen between 1998 and 2014, showing that variations in oceanographic conditions are generally capable of inducing observed variability in thinning rates, while more extreme ocean temperatures than observed in 2014 and additional contributions from changes in surface melt and dynamic thinning may explain the low estimate of the thickness loss based on the melt model alone.

**Consequences of the ice shelf thinning**. A numerical model of the ice flow dynamics was used to assess the buttressing of the floating tongue and the consequences of its further thinning for the grounded ice. For recent conditions the ice shelf is strongly buttressed, especially within the parallel sided main fjord. Only the lateral regions of the frontal part, where the ice shelf expands in the widening fjord, show a less expressed buttressing (Supplementary Fig. 3). This indicates that the glacier remains in a stable condition, even if the frontal part would be removed.

Time dependent experiments, starting from the modern ice geometry (Supplementary Fig. 4), show that a change from equilibrium conditions to a strong negative mass balance (as it is indicated by our results on thinning rates) will have a strong impact on the floating ice, as well as on the adjacent ice sheet (Supplementary Figs. 5, 6). The results for a 100 year forward scenario with a 1.5 times mass balance forcing represent an estimate of the ice shelf/ice sheet evolution for the coming decades. The mean thickness of the ice shelf reduces by about 45% (from 190 m to 75 m) during the modeled 100 years (Supplementary Fig. 5a). However, the proportion of very thin ice

(<10 m) increases from almost 0% to almost 70% (Supplementary Fig. 5b), which indicates instability for the largest part of the ice shelf. A removal of the entire ice shelf in the fjord would lead to a strong thinning of the upstream ice sheet, at least along the 40 km long flowline simulated in our experiments (points 1–7 in Supplementary Fig. 4). At the grounding line the thinning is about 200 m, which leads to a migration of the grounding line by approximately 10 km. Here, the ice thickness reduction reaches 80–100 m after about 40 years, while the flux increases by about 30–40% (not shown here). The thinning still is about 30–50 m between 15 and 20 km upstream of the grounding line. Eventually the ice sheet stabilizes with ice thickness lowered between 100–120 m at the end of the simulation.

**Discussion**

Based on our observations and calculations, 79 North Glacier has lost almost one third of its thickness in the region of the 1998 Midgardsormen experiment between 1998 and 2016. Because the ice shelf is freely floating in the fjord, it can be assumed that this relative thinning is representative of a large part of the ice shelf. Otherwise, the lateral grounding line would not have moved up to shallower bedrock. The results are consistent with previous investigations concerning the bulk mass loss (for example see ref.[1], found a 30% total ice thickness loss downstream of the main grounding line between 1999 and 2014). We can confirm a similar magnitude of ice loss for the central part of the ice shelf and for a similar period. However, our study shows for the first time the temporal pattern in mass wastage for a period of 18 years. The uncertainties in surface elevation derived from remote sensing images and the lack of repeat elevation information make it impossible to infer

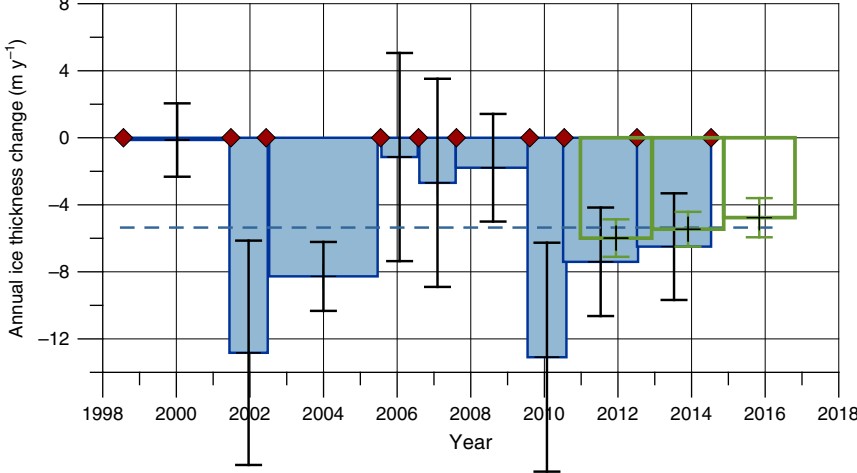

**Fig. 5** Thickness changes of 79 North Glacier based on the migration of Midgardsormen. Temporal evolution of annual mean thickness changes as inferred from the displacement of Midgardsormen between 1998 and 2015. Red dots: date of grounding line detection, blue dashed line: mean annual thickness change between 1998 and 2016. The green boxes represent ice thickness changes from TanDEM-X surface elevation differences derived for recent periods over a larger part of the floating ice shelf. The vertical lines represent the error of the thickness change

the temporal evolution from existing remote sensing information before 2010. However, the combination of bedrock topography and lateral grounding line migration provides the necessary input for deriving such a time series back to 1998. We have shown that the ice shelf experienced a change from a state close to equilibrium to a state of successive thinning, where the interannual variability is large. For equilibrium conditions, the ice thickness remains constant because ice dynamic thickening balances the total melt rates of about 12 m year$^{-1}$ in the region of Midgardsormen, while the thinning doubles during phases of maximum thickness loss in 2002 and 2010.

According to our analysis, there are two periods with very strong mass loss from 2001 until 2005 and from 2009 until 2010. After 2010, the mass loss remains high, but with a decreasing tendency until 2016. Even in the period 2014–2016, the mass loss is almost as high as the mean value of −5.3 m year$^{-1}$ over the entire period 1998–2016. There is no information about the bedrock topography further to the South of Midgardsormen, which would allow the mass budget estimate for the ice ridge migration from 1994 until 1998. Some degree of thickness loss must have happened during this period to facilitate the observed migration of 682 m. This also indicates that the fjord shoulder extends at least this distance further towards the fjord center.

We investigated the potential causes for the observed ice loss, finding that neither a change in ice dynamics, nor a more negative surface mass balance are likely to explain the persistent thinning of the glacier. Instead, we demonstrated that observed variations in ocean temperature at the ice base would induce sufficient additional melting to cause the estimated mass loss of the ice shelf, indicating that the observed thinning relates to changing ocean conditions in front of 79 North Glacier. Warming in the subpolar North Atlantic since the mid-1990s[24–26], a thickening of the AW layer in the Irminger Sea[27] and a warming of AW in the Arctic Ocean in the 2000s[9] have been observed. Warm anomalies in Fram Strait in 1999–2000 and 2005–2007[8], and a shoaling of the AW layer in the eastern Eurasian Basin[28] suggest that these large-scale perturbations may also reach the Greenland coast. Recent studies show a consistent warming and thickening of the AW in eastern Fram Strait and on the North East Greenland continental shelf[29]. Although observations inside the ice shelf cavity are too sparse to scrutinize this trend, the existent

hydrographic profiles suggest a successive warming and thickening of the AW layer that is consistent with the large-scale evolution.

However, the reason for the large interannual fluctuations of the thinning rates revealed by this study still need to be found. AW anomalies in Fram Strait take about 1.5 years and longer to reach the 79 North Glacier[21], while fjord temperatures may vary greatly on shorter time scales. Also the transient adjustment of the cavity circulation further modulates the response of the glacier to ocean forcing[30]. Following the method of[31] and using the hydrographic profiles to constrain the water mass transformation inside the ice shelf cavity (Supplementary Fig. 2b), it can be shown that the observed 40% increase in basal melting is associated with a 30% stronger cavity overturning circulation. Herein, the temperature difference between ingoing and outgoing waters (using the same definitions as ref.[29]) changes little between the different years, but the increased meltwater input at the ice base drives a more vigorous sub-ice shelf circulation that accomplishes the additional heat flux of $0.7 \pm 1.8 \times 10^{11}$ W into the cavity. This results in a reduction of the cavity exchange time scale from about $120 \pm 26$ days to about $90 \pm 35$ days, which further increases the sensitivity of the ice shelf to ocean changes[30]. Thus, while our analysis suggests that the ocean is likely the main driver of the observed changes at 79 North Glacier, the regional dynamics that control the heat transport into the ice shelf cavity and other contributors, such as subglacial discharge induced by surface melt or geothermal heat flux will need further attention to fully understand the observed thickness evolution.

Potential consequences for the future ice shelf stability: Despite the fact that 79 North Glacier has a more stable grounding line situation than Zachariæ Isstrøm (rising bedrock inland), the loss of the ice shelf might contribute to destabilize the entire, marine-based ice sheet sector. At the moment the ice shelf is well buttressed in the fjord and even the loss of the outer part would probably not change this. Without the buttressing effect of the floating ice tongue in the fjord, our simplified model approach demonstrates that the ice thickness strongly decreases and the grounding line retreats by about 10 km. This is comparable to the results of a recent study[32] and poses the question if the disintegration of the ice shelf and its related consequences on the grounded ice are likely to happen in the near future. The ice

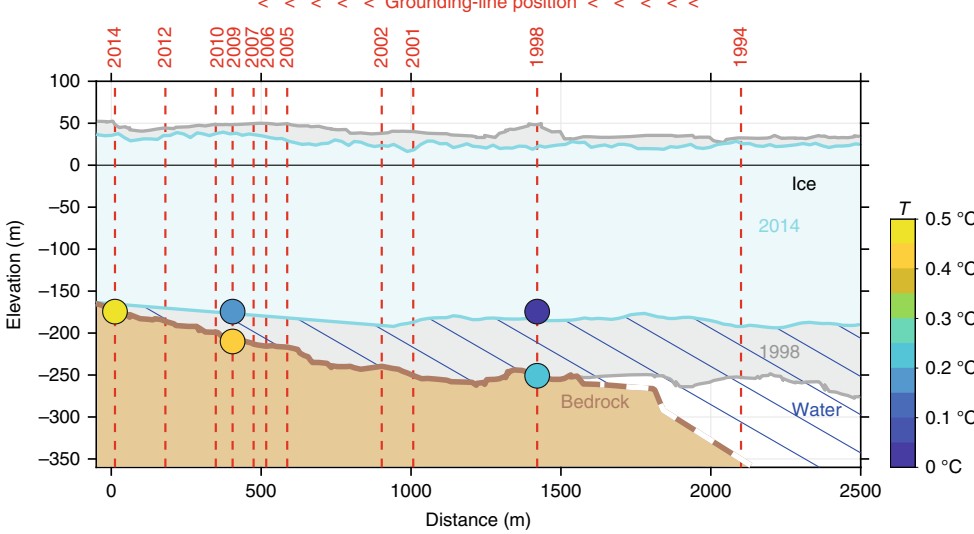

**Fig. 6** Ice shelf geometry and water temperatures at Midgardsormen. Changes in ice shelf geometry in the Midgardsormen region between 1998 and 2014 and the location of the grounding line for the years with suitable satellite imagery. The water temperatures for the years of available CTD measurements are shown for the respective depth levels of the grounding line and a water depth of 175 m (grounding line depth in 2014)

thickness reduced by about 30% during the period 1998 until 2014. Compared with balanced conditions, implying a mass balance of $-12\,\mathrm{m\,year^{-1}}$ in the region of Midgardsormen, the mean ice loss during the observation period results in an almost 1.5 times higher mass balance magnitude. The numerical simulations demonstrate that the thinning will lead to large areas of very thin ice, which are most likely unstable and large parts of the ice shelf will disappear during coming decades. Given that the environmental conditions already enabled ice thickness reductions up to 13 m within one year that process could be considerably faster for enhanced oceanic energy fluxes into the ice shelf cavity. Even though the consequences are serious for the neighboring part of the ice sheet, where the ice thins by about 200 m after the loss of the ice shelf, it seems that the increased fluxes will not reach far into the ice sheet during the next century, resulting in thickness losses in the order of 30 m about 20 km upstream. It needs to be considered, however, that our simple model setup is not appropriate to simulate the long-term feedback mechanisms. Therefore a more detailed investigation is required for investigating the long-term stability of the ice sheet in this sector of Greenland.

## Methods

**In-situ data preparation**. For our analysis, we combine glaciological in situ observations, satellite data, sub ice shelf oceanographic measurements and climatological reanalysis results. The core dataset was collected during joint German-Danish campaigns in 1997 and 1998 within the framework of the project Climate Change and Sea Level (ENV4-CT095-0124), providing values for ice thickness and water depth below the ice shelf at the locations indicated in Fig. 1, based on single shot, 24-channel seismic records[13]. In addition to the regional mapping of the ice shelf geometry, a detailed seismic survey in 1998 across Midgardsormen (Fig. 2b), close to its western origin (center location: 79° 29.94′ N, 22° 17.35′ W), provides the ice thickness and the underlying bedrock elevation (Fig. 3a). In addition, ice velocities were determined by repeat-GPS measurements and a network of tilt meters recorded the spatial pattern of tidal movement of the ice.

These ground-based measurements are complemented by airborne ice thickness measurements that were carried out in 1997, using a low frequency ground penetrating radar system and covering a large part of the ice shelf and the adjacent grounded ice sheet (source: Microwave and Remote Sensing, DTU Space, the Technical University of Denmark). The lines of this data set, which are used in this study, are shown as dark blue lines in Fig. 1.

**Grounding line migration by tracking Midgardsormen**. The temporal evolution of MGO-ridge was tracked on scenes, selected from the Landsat archive, for the

period 1998 until 2015 with roughly annual separation and an additional scene in 1994 (Table 1). For some years, no suitable scene could be identified due to cloud cover during the acquisition times (missing years in Table 1). Displacements of the MGO-ridge were measured manually on the co-registered images with a pixel size of 30 m. The combination of the errors of co-registration of the scenes and the identification of the Midgardsormen position is less than two pixels and thus better than 60 m. In addition, annual ice velocities of the floating ice were determined from surface feature displacements between the Landsat scenes using the IMCORR correlation algorithm[16].

**Errors involved in the ice shelf thickness change**. We estimated the reference surface elevation of the airborne measurements relying on the combination of aircraft GPS positions and travel times of the electromagnetic wave through air (using the first reflection from the ice surface). The accuracy of the ice thickness estimation depends on the center frequency of the radar system and the travel speed of the radar waves in the ice. With the assumption of a wave velocity of $168\,\mathrm{m\,\mu s^{-1}}$ and the system frequency of 50 MHz, the possible vertical resolution (1/4th of the wavelength) of the radar system is about 0.84 m. The final accuracy, however, depends on the difference between the estimated and the wave travel speed in the ice column. Because there is no firn layer on the ice shelf, the radar wave velocity should be about $v = 168 + \text{-}3.4\,\mathrm{m\,\mu s^{-1}}$ (2% error) and thus very close to the theoretical value[33]. A realistic error of determining the two way travel time to the ice/underground reflector is the half wavelength and thus about 1.7 m, or $\varepsilon_\tau = 0.0101\,\mu s$ in travel time. The mean two way travel time in the MGO region was $\tau = 2.4\,\mu s$. Therefore the resulting error in the derived ice thickness is

$$\Delta\varepsilon_{\mathrm{rh}} = \frac{1}{2}\sqrt{\tau^2\varepsilon_v^2 + v^2\varepsilon_\tau^2} = \frac{1}{2}\sqrt{2.4^2 4^2 + 168^2 0.0101^2} = 4.2\,\mathrm{m} \quad (2)$$

according to the ref.[33]).

The GPS positioning error during the airborne survey was about 40 m, which relates to a relative change in ice thickness of about 3.5 m for a mean bed slope of 5°. The resulting ice thickness error from the radar measurements is therefore 5.5 m.

We infer ice thickness change from the floatation condition and thus the bedrock elevation at the identified grounding line position. In order to determine the accuracy of the bedrock elevation, also the surface elevation error of the radar data needs to be included. The surface elevation at the grounding line is derived from the ice thickness and the floatation criterion south of the MGO ridge. The best fit for the free-floating condition at the grounding line results in a residual mean square of the surface elevation of 1.45 m across the entire ice shelf profile. The precision of the surface elevation correction from the radar data to the digital terrain model on the ice-free grounded part of the profile, north of the MGO ridge, is in the order of 2 m. Therefore, the error of the surface elevation along the profile from the MGO ridge to the shoreline is within 2 m.

The error of the bedrock elevation thus results to $\Delta\varepsilon_{\mathrm{b}} = \sqrt{10.4^2 + 2^2} = 5.8\,\mathrm{m}$.

The reconstruction of the ice thickness based on bedrock elevation, MGO position and the floatation conditions thus is affected by a total error of 6.4 m.

**Surface elevation data, errors, and inferred ice thickness.** To assess temporal changes in ice thickness, the airborne measurements from 1997 are compared to more recent smoothed (mean values according to the airborne radar sampling size) surface elevation measurements from the NASA Airborne Topographic Mapper (ATM) (ILATM2_20120514_135706[34]) that were collected in the framework of Operation IceBridge in 2012 and 2014 across 79 North Glacier along an almost identical flight line. The accuracy of the ATM elevation model is given as 2 m. We reduce the absolute error of ATM by calibrating the data over stable ground with the surface reflection of the radar data from 1997.

In addition, TanDEM-X bistatic data acquired on 8-01-2011, 14-11-2012, 8-12-2014, and 28-09-2016 were used for spatially distributed surface elevation information. This data set is characterized by effective baselines ranging from 182 to 75 m with a corresponding height of ambiguity of 38–113 m (details: Supplementary Table 2). The InSAR digital elevation models (DEMs) cover about $30 \times 50$ km$^2$ and were derived using the Integrated TanDEM-X Processor (ITP), the operational interferometric processor of the mission[35]. The absolute height error of the DEMs computed with ITP takes interferometric coherence and geometrical considerations into account[36,37]. As explained in detail in[37] the TanDEM-X global DEM and all intermediate raw InSAR DEMs are affected by the absolute horizontal error, the absolute height error and the relative height error that describes local height variations. In the present study, which relies on TanDEM-X—TanDEM-X raw DEM differencing, we only quantify the absolute height error of each scene separately. The relative height error is estimated as random error together with the final elevation difference measurements. Moreover, the absolute horizontal error is negligible, because of the excellent geolocation accuracy of the ITP processor[37].

Hence, the absolute height error is estimated over ice-free terrain from offsets to the TanDEM-X global DEM. It is different for each raw DEM and during the DEM differencing the respective absolute height errors add up independently to $SE_{\Delta z}$. Together with the statistical error of the elevation difference measurement over the floating ice tongue $SE_{\Delta h}$ the overall uncertainty $\varepsilon_{\Delta h}$ is calculated as:

$$\varepsilon_{\Delta h} = \sqrt{SE_{\Delta z}^2 + SE_{\Delta h}^2} \qquad (3)$$

The uncertainties are reported in Supplementary Table 1. Based on the periods between the acquisitions, ice thickness change rates and their respective errors are reported from buoyancy calculations. Concerning additional errors from signal penetration, the backscattering coefficient $\sigma^0$ has been analyzed over the floating ice tongue (Supplementary Table 2). We find values ranging from approx. $-6$ to $-12$ dB, therefore we assume a dominating surface scattering, because of the crevassed and rough surface.

For the DEMs used in the present study and over the floating part of the glacier the resulting height error is ranging from 0.65 to 1.35 m. The spatial resolution is approximately 12 m. All TanDEM-X RawDEMs are vertically co-registered to the TanDEM-X global DEM over ice-free areas. The error of TanDEM-X—TanDEM-X surface elevation differences over the floating part of the ice tongue is therefore estimated to be better than 0.2 m (Supplementary Table 3), which results in an error of the ice thickness estimate of less than ±2 m.

Finally, we compare measured ice thicknesses from the airborne ground penetrating radar acquisitions in 1997 with surface elevations derived from TanDEM-X elevation models and Operation Ice Bridge ATM profiles. The surface elevation was transferred to ice thickness across the floating ice shelf by using an ice density of 900 kg m$^{-3}$ and an ocean water density of 1028 kg m$^{-3}$.

**Basal melt rates from plume modeling.** The oceanic forcing as a driver of basal melting is assessed through four conductivity temperature depth (CTD) profiles that have been measured inside the ice shelf cavity between 1998 and 2014 during similar seasons (Supplementary Fig. 2b). The first profile was taken in August 1998 through a borehole drilled near the eastern end of the MGO-ridge (Fig. 1, the western CTD location). The other three profiles were taken in close proximity to each other near the northern calving front towards Dijmphna Sund (Fig. 1). Two of these profiles have been accomplished in September 2009[20,31], and one in September 2014[29]. We use an ice-shelf plume model[22] to estimate the sensitivity of basal melt rates to changes in ocean temperature, based on these observations. While the model provides reliable estimates of basal melting for one-dimensional configurations[38], the plume dynamics were augmented to account for a varying width (lateral extent) of the plume perpendicular to the flow direction, which is important to provide area average melt rates for an uneven distribution of ice shelf area at different depths[39]. This is represented in the model by a non-dimensional parameter of change in ice shelf area as function of depth $(dw/dz)$, which was computed by binning and normalizing the gridded ice draft data[40] into 200 m depth bins. To account for the varying geometry of the ice base for different plume paths, a set of five representative profiles of the ice base slope were used (Fig. 1, green lines). The profile lengths vary between 62 and 70 km along the axis of the glacier flow and were chosen to have a regular spacing across the glacier between the grounding line and the calving front (Supplementary Fig. 2a). An ensemble of mean melt rates was computed, by averaging the melt rates along each profile. To estimate the melting sensitivity to changes in ocean temperatures, the model was forced with ambient ocean temperatures provided by the different CTD profiles. Other parameters and constants were adopted from[22], except the entrainment

coefficient, which was set to E0 = 0.016 (as opposed to E0 = 0.036), such that mean melt rates obtained from the 1998 CTD data match with the observed bulk thinning of the glacier.

**Surface mass balance variability.** In order to estimate the variability of the surface mass balance on the ice shelf, we calculated annual positive degree day sums[41] based on air temperature observations at Danmarkshavn located about 300 km south of 79 North Glacier and Station Nord about 200 km to the North (Fig. 1 see ref.[17]). This simple approach will provide the temporal variability of surface melt (Supplementary Fig. 1) and thus is sufficient to estimate the relative surface mass balance changes.

**Ice-dynamic glacier response.** To assess the impact of the observed ice shelf thinning, the open-source 3-D thermomechanically coupled ice-flow model Elmer/Ice[42] was applied to the 79 North Glacier and its upstream region. The stability of the floating ice for present day conditions is evaluated by computing the buttressing field according to the ref.[43]. The future evolution of the glacier system is investigated with a 100 year-long forward simulation, in which we impose a 1.5 times higher (negative) mass balance as is required for present-day steady state conditions. This forcing corresponds to the observed mean thinning rate.

As input data we use surface velocities from feature tracking results[44,45], the TanDEM-X global DEM[37], the ice thickness distribution from from BedMachine Greenland version 3 (see ref.[46], see Supplementary Fig. 4) and a mass balance which is assimilated from ice flux divergence, as explained below. The footprint of the investigated domain is covered by a triangular regular mesh with 1 km spatial resolution.

For the buttressing field, we solve an optimization problem to infer basal friction and stiffening coefficients by matching modeled ice velocities with observed velocities (for example see ref.[47,48]). To avoid overfitting or over-regularization an L-curve analysis was performed to select the optimal parameters for the inversion. From the modeled velocities the 2D buttressing field is computed following[43].

The calculations for the temporal evolution are based on the shallow shelf approximation (SSA, i.e., a 2D representation), using a non-linear constitutive equation (flow exponent $n = 3$) and a linear friction law. The ice viscosity $B$ and the linear friction coefficient $\beta^2$ are estimated deploying standard inverse methods. The defined cost function considers differences with the observed surface velocity field. To avoid flow instabilities during transient runs, the oscillations occurring in the inverted 2D viscosity field are eliminated by defining areas of equal viscosity. The geometry evolution of the glacier is calculated deploying the thickness evolution equation:

$$\frac{\partial H}{\partial t} + \nabla \cdot (\bar{u}H) = a_S + a_b, \qquad (4)$$

where $H$ is the glacier thickness, $t$ the time variable, $\bar{u}$ the mean horizontal velocity, $a_s$ and $a_b$ the surface and bottom mass balance.

The stability of Nioghalvfjerdsfjorden with respect to different states of the terminating ice shelf is investigated running the ice-flow model with a synthetic mass balance scenario. First, a 2D mass balance field $m_s(x, y)$ is estimated using the thickness evolution equation. This is achieved by introducing the velocity field calculated with the flow model in Eq. (4), imposing zero mass balance and retrieving the resulting elevation change $\partial H/\partial t$ distribution. This distribution represents the mass balance necessary to keep the glacier in steady state. Afterwards, the mass balance scenario is modified with respect to the derived thinning rates. As the steady state mass balance is about $-12$ m year$^{-1}$ in the region of Midgardsormen in order to compensate for ice shelf convergence, the mean thinning rate of about $-5.5$ m year$^{-1}$ represents an intensification of the mass balance by roughly a factor of 1.5. This amplifying factor is used for the scenario run, starting from the steady state as initial condition. This is compatible to the scenario of the ref.[32], who increased the basal melt at the grounding line from $-30$ to $-90$ m year$^{-1}$, as our mean melt rates at the grounding line for the scenario run reach about $-100$ m year$^{-1}$.

**Data availability.** The seismic data across Midgardsormen, the CTD profiles and TanDEM-X surface elevation profiles used in this paper are available via the Pangaea data base (https://doi.org/10.1594/PANGAEA.891369, https://doi.org/10.1594/PANGAEA.891386). The airborne radar data are available from the Microwaves and Remote Sensing Division, Danish Technical University (DTU) on request. Meteorological data used for the surface mass balance calculations are available at the Danish Meteorological Institute: http://www.dmi.dk/laer-om/generelt/dmi-publikationer/2013, technical report No. 15-08, John Cappelen (ed.), Weather observations from Greenland 1958–2014—Observation data with description. Landsat data, used for tracking Midgardsormen (Path 11, Row 02), are available from the USGS remote sensing data archive: https://glovis.usgs.gov/app?fullscreen=0. Surface elevation data for 2012 and 2014 are retrieved from the NASA Airborne Topographic Mapper (ATM) data repository: http://nsidc.org/data/ILATM2/versions/2#. The TanDEM global DEM, used for surface elevation information in the numerical modeling experiment, is available on request via a science proposal at the German Aerospace Center (DLR) only. Ice thickness information is taken from BedMachine Greenland version3: http://sites.uci.edu/morlighem/

dataproducts/bedmachine-greenland/, while the surface velocity is used from http://cryoportal.enveo.at/data/ and https://nsidc.org/data/NSIDC-0670/versions/1.

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

## Acknowledgements

The TanDEM-X global DEM tiles and CoSSC were provided by DLR under the research proposals DEM_GLAC0671 and XTI_GLA6663 ©DLR 2017. The work was partly supported by Deutsche Forschungsgemeinschaft (DFG, FL 848/1-1). Surface-elevation measurements were provided by NASA's Airborne Topographic Mapper (ATM) Program.This work was supported in part through grant (OGreen79) from the Deutsche Forschungsgemeinschaft (DFG) as part of the Special Priority Program (SPP)-1889 "Regional Sea Level Change and Society" (SeaLevel). C.S. was supported by the Deutsche Forschungsgemeinschaft (DFG) in the framework of the priority programme "Antarctic Research with comparative investigations in Arctic ice areas" by the grant MA 3347/10-1. The Microwaves and Remote Sensing Division, Danish Technical University (DTU) is acknowledged for providing the airborne ice thickness data.

## Author contributions

C.M. conducted field work, conceived the research and wrote the article; J.S. and T.H. analysed the oceanographic data and performed the plume modeling; D.F. and L.K. processed and analysed the TanDEM-X data; P.A.D. provided CTD measurements and input to the oceanographic discussion; T.K. discussed the oceanographic part and

analysed the melt impact; C.S. and C.L. calculated the buttressing and performed the ice-dynamic model experiments. All authors contributed to the writing of the manuscript and the revisions.

## Additional information

**Competing interests:** The authors declare no competing interests.

