## [Peer Review File · Nature Communications]

Reviewers' comments:

Reviewer #1 (Remarks to the Author):

Major concerns:

(1) Overall, the paper is very well written and well organized. The main message of the study is that high values of up to 13 m ice thickness reduction within one year indicate that the ice shelf could lose up to 75% of its thickness within one decade for given environmental conditions. My main concern is that the paper focus on a small area of 79 glacier, and do not show why (supported by evidence) this is important for the mass balance of the northeast Greenland ice stream (NEGIS). To publish in nature communications, the study must deliver something more than just an improvement of the temporal evolution of bulk of ice loss on 79 glacier. The study could include ice flow modeling (see e.g. Choi et al, 2017) and show how the observations presented in this study could have an impact on the total mass budget of the drainage basin. A paper that simply improve the temporal evolution of the floating part of the 79 glacier is more suitable for a technical journal.

(2) Line 365 to 376 list potential consequences of breakup of the floating ice tongue in the fjord. Again, my concern is that none of these consequences are supported by any analyze presented in the paper. As far as I see it, this entire section is speculations, and should therefore be removed. I suggest the authors either delete the section or include ice flow modeling that potentially could support their statements. Modeling is important, for instance, the Peterman glacier in northwest Greenland lost a huge amount of the floating tongue in 2012, and however, it did not generate any notable ice loss upstream glacier. Similar, if you claim that breakup of the floating ice tongue will have consequences for the upstream area; this must somehow be demonstrated in the paper.

Minor comments

I think the title is misleading, "causes" are not showed in any way in this paper. Also, by "Nioghalvfjordsjorden Glacier", I imagine the entire glacier and especially the grounded portion of the glacier. However, this paper focus on the floating tongue, which if melted completely will not cause sea level rise (only melt of land based ice causes sea level rise).

Line 12: "Nioghalvfjordsjorden is one of the largest outlet glaciers", in term of what ? drainage area? flux rate ? ice loss?

Line 13. "A considerable loss in ice thickness was observed across the floating part of this glacier since 1999". Well, floating ice do not contribute to sea level rise. So why is this interesting?

Line 27: "1.1 m global sea level rise". Out of context. This paper is not about sea level rise!

Line 33: "The area of 79 North Glacier has remained remarkably stable since about 1906 (7)". I find this statement confusing. Reference 7 was published in 1913, and discuss the area during 1909-1912. How can this reference justify stability during 1913-present!

Choi, Y., Morlighem, M., Rignot, E., Mouginot, J., & Wood, M. (2017). Modeling the response of Nioghalverdsorden and Zachariae Isstrøm glaciers, Greenland, to ocean forcing over the next century. *Geophysical Research Letters*, 44, 11,071–11,079.
<https://doi.org/10.1002/2017GL075174>

Reviewer #2 (Remarks to the Author):

Summary

This manuscript presents an improved time series of ice shelf thinning rates from NE Greenland in

a region important for solid ice discharge from the ice sheet, and freshwater delivery to the Nordic Seas. Existing temporally-sparse remotely-sensed, and field data are supplemented by thickness estimates based on tracking the surface expression of a lateral grounding line. These innovative data reveal highly variable mean annual ice shelf thinning rates, which, due to their high magnitude, implicate increased ocean heat flux as the only plausible cause. If the greatest observed thinning rates were sustained, the ice shelf could lose 75 % of its thickness over a single decade, with significant implications for upstream ice flow and calving rates. This is an important and interesting paper that I believe will be of great interest to others in the community and the wider field. My comments are mostly fairly minor and concern grammar and wording, but there are quite a few of these issues that I believe need to be addressed.

Specific points (by Line number, L)

L12: Given the next sentence, I wonder if it is worth expanding on how this observation was made.

L13: Consider changing 'was' to 'has been'.

L13: 'Lack in' should be 'Lack of'.

L14: Consider changing '...no temporal evolution of this bulk ice loss has been presented so far...' to '...to date no temporal evolution of this bulk ice loss has been presented...'.

L15-16: Consider changing '...ice thickness and bedrock data it is possible to describe...' to '...ice thickness and bedrock data, we describe...'.

L18: Consider adding 'instead' between 'are' and 'governed'.

L19-20: Consider changing 'The high values of up to 13 m ice thickness reduction within one year indicate...' to 'Observed thinning of up to 13 m/yr indicates...'.

L23: 'of' should be 'in'.

L24: 'at its centre' sounds a bit odd and I wonder if 'at mid-distance' might be better?

L30: 'early' should be 'already'.

L30-38: How does this paragraph equate with the second sentence of the abstract? It would be worth checking that both parts are consistent.

L35: Consider inserting 'has' before 'experienced'.

L36: Missing a full-stop.

L37: In some places it is 'the 79...' and in others just '79...'. I would pick one and be consistent (I prefer no 'the'). Also missing an apostrophe 'Glacier's'.

L39: Consider changing 'Recent observations of' to 'Recently observed'. Otherwise the sentence suggests that it is the observations that will affect the stability, rather than the changes themselves.

L43-44: These areas should be annotated on Figure 1 given that they are mentioned here.

L44: Insert 'have' between 'measurements' and 'revealed'

L45: Consider changing 'It's deepest' to 'The glacier's thickest'.

L48: 'Represents a remarkable'

L53-55: I wonder if it the best approach to present the final conclusions so early on.

L53-54: Consider changing 'data on oceanic and atmospheric forcing' to 'oceanic and atmospheric forcing data'.

L74: 'providing values for ice thickness'. Also, remove 'the' before 'water depth'.

L78: Consider changing 'provides the ice thickness and the bedrock elevation underneath' to 'provides the underlying ice thickness and the bedrock elevation'.

L86-87: The grey shading is not clear in the pdf reproduction of the figure.

L88: Consider changing 'Those' to 'These'.

L106-107: Consider changing 'Landsat archive have been selected for' to 'Landsat archive for'.

L107: Consider changing 'periods' to 'separation'

L115: Consider changing 'The oceanic' to 'Oceanic'

L116: Consider changing 'have been' to 'were'

L119: 'calving front towards'

L120-121: Consider changing 'An ice-shelf plume model (20) estimates' to 'We employ an ice-shelf plume model (20) to estimate'

L114: Consider changing 'So far' to 'To date' (otherwise could be misconstrued as meaning within this article only).

L145-146: Consider changing 'The comparison of the ice thicknesses on the ice shelf from' to 'Comparison of ice shelf thicknesses from'

L152: I would recommend being consistent with the phrasing and sign of the thickness changes.

L157: 'with' should be 'as'.

L167: 'high temporal resolution' is vague. It would be better to give an actual value, for example 'an approximately annual temporal resolution'

L168: Consider replacing 'the ice ridge feature' with 'it'.

L173: Make clear that it is the lower panel of Figure 3 that is referred to here.

L189-191: It would be better to have consistent axes extent (i.e. figure panel extent) and font sizes for the two parts of this figure.

L193: It would be good to annotate the first 3.2 km of the light blue line on Figure 4.

L195: To be consistent you should also state the relevant vertical exaggeration value for the upper figure.

L197: I realise it may seem obvious, but it would aid in quick interpretation of the figure to add 'N' and 'S' at the ends of the profile presented in Fig. 3.

L214: Consider changing 'grounding line. This inhibits' to 'grounding line, which inhibits'.

L220-221: Change the ',' in column 3 of the table to '.' to be consistent with the main text.

L224: I don't think that 'will' is necessary here.

L257: Space after 'until'.

L281 (and elsewhere): 'Atlantic Water'

L317: The modelled melt rate is comparable but quite a bit (~30 %) lower. Any ideas why? Did the plume consist only of melted ice shelf - i.e. was there any additional 'forced' convection based on the subsurface runoff of geothermal melt and basal frictional melt at the grounding line? The inclusion of realistic values for these may act to increase the model-derived melt rates. I don't think it is necessary to re-run the model, but I think it would be a good idea to at least mention some reasons to explain the relatively low modelled melt rates.

L336: Consider replacing 'this' with 'our', otherwise the meaning is slightly ambiguous as you could also be referring to reference (1).

L341: Change 'could show' to 'have shown'.

L357: Consider changing 'We could demonstrate that the ice loss into the ocean water below' to 'We have demonstrated that basal ice melt by ocean water below'

L363: 'for' should be 'in'

L363-364: What about increased surface melt and/or increased basal melt? Maybe not from ice acceleration, but perhaps from temporal variations in geothermal heat flux and atmospheric temperatures? Surface melt was higher 2001-2005 and 2009-2010 (Figure S1). This might be worth a brief discussion.

L366: 'towards the' is unnecessary.

L371: Consider changing 'ends' to 'results'.

L373: 'loses' could change to 'could lose'.

L374: Not just the rate of entrainment (presumably related to the volume of subsurface glacier meltwater runoff at the grounding line?), but the water temperature too. Maybe: 'sustained high sub ice shelf oceanic heat flux' would be better than 'intensified warm water entrainment'?

L375: Consider adding 'with' after 'However,'.

Reviewer #3 (Remarks to the Author):

- Key results: This is an interesting paper about thickness changes on a major glacier in Greenland. The authors use a combination of in situ and remote sensing observations, combined with oceanographic measurements to conclude that ocean-driven basal melting has caused the long-term changes in ice thickness. The strength of this paper is the fact that the authors have a new result (quantification of thickness change) and some creative methodology (using the

migration of a shear zone to derive long-term thickness changes).

- **Validity:** The main conclusion, that ice shelf thinning is due to basal melting from warming ocean temperatures, is essentially based on 4 CTD casts taken years/decades apart. There is an abundance of literature showing that fjord temperatures undergo large seasonal changes, so inferring anything from a few point measurements is tenuous. I recognize the modeling work that the authors did to combat the data scarcity, but am still skeptical.

- **Originality and significance:** The use of a shear margin to infer thickness change is original and the high rates of thinning on this ice shelf are definitely interesting and significant. However, as it is written now, I do not find this paper to be of "immediate interest" to non-glaciologists.

- **Data & methodology:** This work uses a lot of very disparate datasets (ground-based, remote sensing and modeling). While I find the writing and organization hard to follow, the authors do include all the relevant data descriptions.

- **Appropriate use of statistics and treatment of uncertainties:** Yes, the authors are careful about statistics.

- **Conclusions:** Overall, I found the conclusion that the ice shelf has thinned to be convincing and well documented. The inferences about atmospheric forcing from positive degree day estimates and a 20 year old plume model based on 4 CTD casts are not very convincing (or as well described).

- **Clarity and context:** This paper is possibly Nature-worthy if it was easier to follow and written more concisely. The first time I read the paper I thought the shear-zone analysis was going to be the major point of the paper. But, the main conclusion is the long-term thinning (which is an interesting result), and the shear zone is just one tool used to derive thickness change.

I've included some specific comments for the first few pages. However, most of these are editorial comments, so I did not continue to make the corrections for the latter part of the text. Throughout the text, the verb tenses are confusing, there are multiple typos and it doesn't seem to follow the Nature guidelines (for length, location of methodology, structure of abstract, or placement of figures).

Abstract: It is slightly confusing what the main question/problem is here. For example, "A considerable loss in ice thickness was observed" – is this your result or a previous observation? Per Nature guidelines, they like the summary paragraph to state the problem and then conclude with "Here we show..."

15: The transition from "Based on the migration of a surface feature" to "ice thickness and bedrock data" is awkward (and missing some verbs).

17: "for producing" -> to produce

18: This statement seems overly confident...your results definitely suggest this conclusion, but not definitively.

27: This sentence needs more detail – could contribute to 1.1 m of sea level rise (under what conditions? Over what time interval? Based on what?)

28: Add some more detail to this implication – how much does it contribute to freshwater flux? A significant amount? It's a pretty slow moving glacier that does not calve icebergs frequently.

30: early?

31: Need references for this statement

36. Missing a period

37: "strong increase in ice flux" – by how much?

38: Glacier's

39-42: This paragraph doesn't seem necessary, especially given the strict word limit

44: "extensive cavity beneath the ice shelf" – does this just mean that the shelf is floating, or that there is a big bed depression under the ice shelf?

48: represents "a" remarkable...

53: The transition to this last sentence is awkward

Fig 1: Suggest changing the color of one of the "light blue" features and adding the location of the grounding line.

60: as "a" red dot

94: How were the ATM data smoothed? Why?

106: "was tracked" and "have been selected" ?

We carefully revised the manuscript and tracked the changes. All remarks of the reviewers are listed and commented below.

Reviewers' comments:

Reviewer #1 (Remarks to the Author):

Major concerns:

(1) Overall, the paper is very well written and well organized. The main message of the study is that high values of up to 13 m ice thickness reduction within one year indicate that the ice shelf could lose up to 75% of its thickness within one decade for given environmental conditions. My main concern is that the paper focus on a small area of 79 glacier, and do not show why (supported by evidence) this is important for the mass balance of the northeast Greenland ice stream (NEGIS). To publish in nature communications, the study must deliver something more than just an improvement of the temporal evolution of bulk of ice loss on 79 glacier. The study could include ice flow modeling (see e.g. Choi et al, 2017) and show how the observations presented in this study could have an impact on the total mass budget of the drainage basin. A paper that simply improve the temporal evolution of the floating part of the 79 glacier is more suitable for a technical journal.

We would like to emphasize that this the first time that detailed thinning rates could be derived for quite a long period on a major outlet glaciers of NE Greenland. There is no other way to retrieve such a unique data set for the past, than by the observation of the Midgardsormen ice ridge. This has never been demonstrated before and the variability is surprisingly large. Therefore, we think that this is a major contribution to the discussion of the mass balance conditions in NE Greenland.

However, we agree with the reviewer that a broader context will definitely strengthen the impact of the manuscript. For this purpose we conducted a numerical simulation of ice flow for this sector of the ice sheet. Due to time constraints and because this is not the focus of the manuscript, we kept the model setup simple, but robust with regards to the envisaged dynamic consequences. The model itself is now briefly described in the Supplementary Material, including the results of the simulation. In the manuscript itself we refer to this material in the methods section and discuss the results in the Results and Discussion, also in relation to the results of Choi et al. (2017). The outcome is that the floating part of the glacier very likely will disappear during the coming decades and that there is a considerable but confined influence on the ice flow of the adjacent ice sheet.

(2) Line 365 to 376 list potential consequences of breakup of the floating ice tongue in the fjord. Again, my concern is that none of these consequences are supported by any analyze presented in the paper. As far as I see it, this entire section is speculations, and should therefore be removed. I suggest the authors either delete the section or include ice flow modeling that potentially could support their statements. Modeling is important, for instance, the Peterman glacier in northwest Greenland lost a huge amount of the floating tongue in 2012, and however, it did not generate any notable ice loss upstream glacier. Similar, if you claim that breakup of the floating ice tongue will have consequences for the upstream area; this must somehow be demonstrated in the paper.

We now performed several model runs for investigating the actual ice shelf stability and the potential break-up consequences. The set-up of a fully-fledged ice shelf break-up model was

not in the scope of our revision. However, we investigated the temporal thinning of the ice shelf. From these results it is very likely (a major part of the ice shelf reduces to ice thicknesses below 10 m within the simulated period of 100 years) that the ice shelf will disintegrate during the coming decades. We now support our description by the model results.

Minor comments

I think the title is misleading, “causes” are not showed in any way in this paper. Also, by “Nioghalvfjærdsfjorden Glacier”, I imagine the entire glacier and especially the grounded portion of the glacier. However, this paper focus on the floating tongue, which if melted completely will not cause sea level rise (only melt of land based ice causes sea level rise).

We removed "causes" from the title. There is no defined upper boundary of Nioghalvfjærdsfjorden Glacier. The name is usually used for the floating tongue alone, as the name “fjord” indicates. Therefore we leave it as it is.

Line 12: “Nioghalvfjærdsfjorden is one of the largest outlet glaciers”, in term of what ? drainage area? flux rate ? ice loss?

This refers to drainage area and ice flux. However, there is not enough space in the abstract to be more specific. We now describe it as a major outlet glacier.

Line 13. “A considerable loss in ice thickness was observed across the floating part of this glacier since 1999”. Well, floating ice do not contribute to sea level rise. So why is this interesting?

We do not understand this question. Are only observations interesting, which influence the global sea level? If environmental conditions lead to a massive loss of the total ice volume, this is a serious consequence in itself. Even more so, because this ice shelf represents the drainage channel for a significant part of the ice sheet. The potential consequences are now presented in a more concise way in the discussion.

Line 27: “1.1 m global sea level rise”. Out of context. This paper is not about sea level rise!

We agree that this paper does not deal with sea level rise. However, it is important to put the work into the context that variations in the ice shelf will influence the dynamic conditions of the ice drainage in the region and thus the ice resources of this sector of the ice sheet. Therefore we prefer to keep the description about the significance of this region of the ice sheet.

Line 33: “The area of 79 North Glacier has remained remarkably stable since about 1906 (7)”. I find this statement confusing. Reference 7 was published in 1913, and discuss the area during 1909-1912. How can this reference justify stability during 1913-present!

In Einar Mikkelsens account the observations of the group of Mylius Erichsen in 1906 are described, according to the sketch maps of Høeg Hagen. All following expeditions found the calving front at more or less the same position between the ice rumpled at about 19.5° W. We rewrote the sentence, to make it clear that this was just the first observation.

Reviewer #2 (Remarks to the Author):

Summary

This manuscript presents an improved time series of ice shelf thinning rates from NE Greenland in a region important for solid ice discharge from the ice sheet, and freshwater delivery to the Nordic Seas. Existing temporally-sparse remotely-sensed, and field data are supplemented by thickness estimates based on tracking the surface expression of a lateral grounding line. These innovative data reveal highly variable mean annual ice shelf thinning rates, which, due to their high magnitude, implicate increased ocean heat flux as the only plausible cause. If the greatest observed thinning rates were sustained, the ice shelf could lose 75 % of its thickness over a single decade, with significant implications for upstream ice flow and calving rates. This is an important and interesting paper that I believe will be of great interest to others in the community and the wider field. My comments are mostly fairly minor and concern grammar and wording, but there are quite a few of these issues that I believe need to be addressed.

Specific points (by Line number, L)

L12: Given the next sentence, I wonder if it is worth expanding on how this observation was made.

We included: "by comparison of digital elevation models". However, we also had to shorten the abstract.

L13: Consider changing 'was' to 'has been'.

Text has changed

L13: 'Lack in' should be 'Lack of'.

Done

L14: Consider changing '...no temporal evolution of this bulk ice loss has been presented so far...' to '...to date no temporal evolution of this bulk ice loss has been presented...'.

Done

L15-16: Consider changing '...ice thickness and bedrock data it is possible to describe...' to '...ice thickness and bedrock data, we describe...'.

Done

L18: Consider adding 'instead' between 'are' and 'governed'.

Done

L19-20: Consider changing 'The high values of up to 13 m ice thickness reduction within one year indicate...' to 'Observed thinning of up to 13 m/yr indicates...'.

Done

L23: 'of' should be 'in'.

Done

L24: 'at its centre' sounds a bit odd and I wonder if 'at mid-distance' might be better?

Substituted

L30: 'early' should be 'already'.

Done

L30-38: How does this paragraph equate with the second sentence of the abstract? It would be worth checking that both parts are consistent.

The two parts do not contradict each other. In the abstract we state that due to a lack of observations it is not possible to quantify the temporal evolution of mass change of 79 North Glacier. In the paragraph of the Introduction we explain that the spatial extent of the glacier has been stable since the first observations. However, the disintegration of the frontal shelf of Zachariæ Isstrøm poses the question about the stability of the 79 North Glacier's floating part.

L35: Consider inserting 'has' before 'experienced'.

Done

L36: Missing a full-stop.

Corrected

L37: In some places it is 'the 79...' and in others just '79...'. I would pick one and be consistent (I prefer no 'the'). Also missing an apostrophe 'Glacier's'.

We now use "79 North Glacier" throughout the manuscript.

L39: Consider changing 'Recent observations of' to 'Recently observed'. Otherwise the sentence suggests that it is the observations that will affect the stability, rather than the changes themselves.

Changed

L43-44: These areas should be annotated on Figure 1 given that they are mentioned here.

Done

L44: Insert 'have' between 'measurements' and 'revealed'

Done

L45: Consider changing 'It's deepest' to 'The glacier's thickest'.

Here we refer to the cavity, not the glacier. Therefore “deepest” should be appropriate.

L48: ‘Represents a remarkable’

Inserted

L53-55: I wonder if it the best approach to present the final conclusions so early on.

We removed this sentence and now only mention the analysis of possible drivers.

L53-54: Consider changing ‘data on oceanic and atmospheric forcing’ to ‘oceanic and atmospheric forcing data’.

Done

L74: ‘providing values for ice thickness’. Also, remove ‘the’ before ‘water depth’.

Changed

L78: Consider changing ‘provides the ice thickness and the bedrock elevation underneath’ to ‘provides the underlying ice thickness and the bedrock elevation’.

Because it is only the bedrock which is underlying the ice, we now wrote: “provides the ice thickness and the underlying bedrock elevation”.

L86-87: The grey shading is not clear in the pdf reproduction of the figure.

We intensified the grey shading.

L88: Consider changing ‘Those’ to ‘These’.

Done

L106-107: Consider changing ‘Landsat archive have been selected for’ to ‘Landsat archive for’.

The sentence reads now: “The temporal evolution of MGO-ridge was tracked on scenes, selected from the Landsat archive, for the period 1998 until 2015...”

L107: Consider changing ‘periods’ to ‘separation’

Changed

L115: Consider changing ‘The oceanic’ to ‘Oceanic’

Not changed

L116: Consider changing ‘have been’ to ‘were’

Changed

L119: 'calving front towards'

Inserted

L120-121: Consider changing 'An ice-shelf plume model (20) estimates' to 'We employ an ice-shelf plume model (20) to estimate'

Changed

L114: Consider changing 'So far' to 'To date' (otherwise could be misconstrued as meaning within this article only).

Changed

L145-146: Consider changing 'The comparison of the ice thicknesses on the ice shelf from' to 'Comparison of ice shelf thicknesses from'

Changed

L152: I would recommend being consistent with the phrasing and sign of the thickness changes.

We tried to be consistent now and describe always the loss or loss rate as a positive value.

L157: 'with' should be 'as'.

Corrected

L167: 'high temporal resolution' is vague. It would be better to give an actual value, for example 'an approximately annual temporal resolution'

Changed

L168: Consider replacing 'the ice ridge feature' with 'it'.

Replaced

L173: Make clear that it is the lower panel of Figure 3 that is referred to here.

Thank you for pointing at a given but wrong distance, which was left from an earlier version of the figure. We corrected this now. The distance is the same for the upper and the lower panel in Fig. 3.

L189-191: It would be better to have consistent axes extent (i.e. figure panel extent) and font sizes for the two parts of this figure.

The figure compares the seismic measurements (upper panel) with the airborne radar measurements (lower panel), but with the same distance reference. We now used the same panel size and font size.

L193: It would be good to annotate the first 3.2 km of the light blue line on Figure 4.

We annotated the light blue line now in the caption of Fig. 4.

L195: To be consistent you should also state the relevant vertical exaggeration value for the upper figure.

Done

L197: I realise it may seem obvious, but it would aid in quick interpretation of the figure to add 'N' and 'S' at the ends of the profile presented in Fig. 3.

We now explained the light blue line with respect to Fig. 3. Therefore it is not necessary to include "N" and "S" in addition.

L214: Consider changing 'grounding line. This inhibits' to 'grounding line, which inhibits'.

Done

L220-221: Change the ',' in column 3 of the table to '.' to be consistent with the main text.

Changed

L224: I don't think that 'will' is necessary here.

Removed

L257: Space after 'until'.

Inserted

L281 (and elsewhere): 'Atlantic Water'

Changed

L317: The modelled melt rate is comparable but quite a bit (~30 %) lower. Any ideas why? Did the plume consist only of melted ice shelf - i.e. was there any additional 'forced' convection based on the subsurface runoff of geothermal melt and basal frictional melt at the grounding line? The inclusion of realistic values for these may act to increase the model-derived melt rates. I don't think it is necessary to re-run the model, but I think it would be a good idea to at least mention some reasons to explain the relatively low modelled melt rates.

In general modelled melt rates are comparable to observations but the computed ice thickness reduction based on the increase in modelled melt rates is 30% lower than suggested by the retreat of Midgardsormen ridge. The modelled average melt rates using, e.g., 1998 hydrographic conditions (8.7 ± 1.1 m/yr) compare nicely to glacier mass budget calculations from that time (8 m/yr, Mayer et al (2000, GRL)). Furthermore maximum melt rates of 40-60 m/yr at 5-10 km downstream the grounding line are inferred from the model and comparable to melt rates published by Wilson et al. (2017). Thus, we believe that the model represents the ice-ocean interaction quite well. Using the model we estimated the ice thickness loss over time

(i.e., taking also into account the ice velocity) having in mind that the uncertainties are relatively large. The total thinning based on glacier observation is clearly at the upper bound of our ice loss estimate from the model. However, smaller contributions from surface melt and/or changes in the ice flux most likely also contribute to the overall ice loss.

You are right that we initialize the model by using typical values for the meltwater flux beneath ice streams ($1 \times 10^{-3} \text{ m}^2/\text{s}$). Subsurface runoff from geothermal flux melt and basal frictional melt is however expected to be negligible, while subglacial discharge originating from surface melt which drains to the bed of the ice sheet is likely to play an important role for an additional "forced" convection. However, the initial flux of subglacial runoff for glaciers around Greenland is not well constrained and most likely highly variable in space and time (e.g. Straneo et al., 2011). In general, subglacial runoff may accelerate ice shelf basal melting near the grounding line in summer (e.g. Motyka et al., 2003, 2011; Straneo and Heimbach, 2013). By freshening the meltwater plume and thus increasing the initial density contrast to ambient water, subglacial runoff enhances basal melting close to the grounding line. Higher ice-shelf basal melt rates are expected in warmer summers, which may cause large seasonal and interannual variability in basal melt rates. Further model experiments applied to 79N Glacier suggest that a change in the freshwater discharge by four orders of magnitude increases the melting by 50%, i.e., increasing the average basal melt rate by about 5 m/yr^{-1} .

While we agree with the reviewer that this issue merits discussion in the manuscript, we believe that thoroughly revising the above points would be slightly misleading. Instead, we added the following sentences to clarify the interpretation of the model results, also having in mind the comments of the reviewer on the temporal variability of the oceanic forcing.

"The results yield a total ensemble-mean ocean induced ice thickness loss of $61 \pm 20 \text{ m}$. The estimate is comparable to the observed thinning of 89 m at Midgardsormen between 1998 and 2014, showing that variations in oceanographic conditions are generally capable of inducing observed variability in thinning rates, while more extreme ocean temperatures than observed in 2014 and additional contributions from changes in surface melt and dynamic thinning may explain the slight underestimate of the thickness loss based on the melt model alone."

L336: Consider replacing 'this' with 'our', otherwise the meaning is slightly ambiguous as you could also be referring to reference (1).

Replaced

L341: Change 'could show' to 'have shown'.

Changed

L357: Consider changing 'We could demonstrate that the ice loss into the ocean water below' to 'We have demonstrated that basal ice melt by ocean water below'

Changed

L363: 'for' should be 'in'

Changed

L363-364: What about increased surface melt and/or increased basal melt? Maybe not from ice acceleration, but perhaps from temporal variations in geothermal heat flux and atmospheric temperatures? Surface melt was higher 2001-2005 and 2009-2010 (Figure S1). This might be worth a brief discussion.

While geothermal heat flux is unlikely to account for large subglacial discharge, variability in surface melt may be relevant for the observed ice thickness variability not only due to surface mass loss but also due to changes in subglacial discharge. From Figure S1 we find increased surface melt in 2002-2005, 2008, and 2011-2014. This may be linked to the increased ice loss in 2002-2005 but cannot explain the strong ice loss observed in 2010.

Also in response to the comments of reviewer #3, we rewrote the discussion in the above lines, which now provides more detailed information on oceanic observations of a warming/shoaling of the Atlantic water layer and includes a discussion of the above aspects:

"We investigated the potential causes for the observed ice loss, finding that neither a change in ice dynamics, nor a more negative surface mass balance are likely to explain the persistent thinning of the glacier. Instead, we demonstrated that observed variations in ocean temperature at the ice base would induce sufficient additional melting to cause the estimated mass loss of the ice shelf. [...] While our analysis suggests that the ocean is likely the main driver of the observed changes at 79 North Glacier, the regional dynamics that control the heat transport into the ice shelf cavity and other contributors, such as subglacial discharge induced by surface melt or geothermal heat flux will need further attention to fully understand the observed thickness evolution."

L366: 'towards the' is unnecessary.

Removed

L371: Consider changing 'ends' to 'results'.

Changed

L373: 'loses' could change to 'could lose'.

Changed

L374: Not just the rate of entrainment (presumably related to the volume of subsurface glacier meltwater runoff at the grounding line?), but the water temperature too. Maybe: 'sustained high sub ice shelf oceanic heat flux' would be better than 'intensified warm water entrainment'?

Formulation changed to: "in case of sustained high ocean heat flux into the ice shelf cavity."

L375: Consider adding 'with' after 'However,'.

Done

Reviewer #3 (Remarks to the Author):

- **Key results:** This is an interesting paper about thickness changes on a major glacier in Greenland. The authors use a combination of in situ and remote sensing observations, combined with oceanographic measurements to conclude that ocean-driven basal melting has caused the long-term changes in ice thickness. The strength of this paper is the fact that the authors have a new result (quantification of thickness change) and some creative methodology (using the migration of a shear zone to derive long-term thickness changes).
- **Validity:** The main conclusion, that ice shelf thinning is due to basal melting from warming ocean temperatures, is essentially based on 4 CTD casts taken years/decades apart. There is an abundance of literature showing that fjord temperatures undergo large seasonal changes, so inferring anything from a few point measurements is tenuous. I recognize the modeling work that the authors did to combat the data scarcity, but am still skeptical.

We fully agree with the reviewer that the few observations beneath the ice shelf cannot be used to infer a warming of ocean temperatures inside the cavity. This was never our intention and the statement that the observed ice shelf thinning is likely to be primarily driven by oceanic changes was not derived from the 4 CTD casts on their own. Instead, this main conclusion is essentially based on the combination of the findings that:

- 1. Surface melt and changes in ice dynamics can most likely be ruled out to have caused the observed thickness changes (exclusion of other drivers).*
- 2. Unlike the atmospheric and ice dynamical changes, the observed variations in ocean temperature inside the cavity are indeed capable of inducing changes in basal melting that are large enough to cause the estimated mass loss of the ice shelf (plausibility of the mechanism). This inference is robust independent of whether the observed temperature changes are part of a seasonal cycle (all profiles are from August and September, suggesting limited influence of seasonal changes) or due to a successive warming signal (which is consistent with, but not proven by the data).*
- 3. Recent literature shows that the Atlantic Water layer off the NE Greenland coast and on the shelf has coherently been warming and shoaling over the period when the glacier thinning occurred, implying that some of that signal may also propagate further into ice shelf cavity (Consistency with large scale trends). In particular, we compared the few CTD casts taken in the sub-ice cavity with observations from the Northeast Greenland continental shelf (where we find a large number of hydrographic profiles taken between 1984 and recent years, Schaffer et al., 2017) and the Fram Strait. This material is currently prepared as part of a separate publications and shows that the cavity profiles reflect the overall warming trend found on the continental shelf and in the East Greenland current.*

While we are confident that this conclusion is a relevant contribution and based on robust analyses, we acknowledge that the argumentation may not have been clear enough in the previous version of the manuscript. Thus, we have revised the abstract, results and conclusions section to be more differentiated. Particular changes were made,

Changes in the abstract:

"Changes in ice flux and surface ablation are too small for producing the temporal variability in thickness evolution which are instead governed by the interaction between the ice shelf and the underlying ocean."

To:

"While changes in ice flux and surface ablation are too small to produce the temporal variability in thickness evolution, increased ocean heat flux is the most plausible cause of the observed thinning."

Changes in the results:

"The results yield a total ensemble-mean ocean induced ice thickness loss of 61 ± 20 m, a figure that is comparable to the observed thinning of 89 m at Midgardsormen between 1998 and 2014"

To:

"The results yield a total ensemble-mean ocean induced ice thickness loss of 61 ± 20 m. The estimate is comparable to the observed thinning of 89 m at Midgardsormen between 1998 and 2014, showing that variations in oceanographic conditions are generally capable of inducing observed variability in thinning rates, while more extreme ocean temperatures than observed in 2014 and additional contributions from changes in surface melt and dynamic thinning may explain the slight underestimate of the thickness loss based on the melt model alone."

The discussion has been extended by adding more detailed information and references on oceanic observations of a warming/shoaling of the Atlantic water layer, as well as by adding a discussion of ocean heat fluxes into the cavity, and emphasizing lack of hydrographic observations from the 79 North Glacier cavity.

• Originality and significance: The use of a shear margin to infer thickness change is original and the high rates of thinning on this ice shelf are definitely interesting and significant. However, as it is written now, I do not find this paper to be of "immediate interest" to non-glaciologists.

The findings in this manuscript have significant impact on the way NE Greenland is regarded in the context of climatic variability. Until about 10 years ago, the ice sheet in this region was assumed stable or even slightly growing. Now it becomes more and more obvious that also this region is affected by environmental changes with a probably increasing intensity. Still, especially 79 North Glacier seemed to be close to balance with an unchanging geometry. Even though it was obvious from surface elevation measurements that the ice shelf undergoes a considerable thinning, only our new results indicate that the variability of ice thickness reduction is enormous. Our investigations indicate that only the ocean conditions can provide enough energy for such high ice loss rates, but no final mechanism is clearly identified. Therefore, this paper will be of high significance for the oceanographic community which need to come up with ideas how the energy transfer can be accomplished and how the effective energy exchange works across the shallow shelf region and into the sub-ice cavity. On the other hand, it also becomes obvious that the ice shelf itself is much more vulnerable than expected before. Our simple model runs show that for current conditions the ice shelf is prone to disappear within the next decades. Even though it is not the focus of this manuscript,

our numerical experiments indicate that such a loss would have a severe impact on the upstream ice sheet flux. Therefore, also the glacier and climate community will have to consider such an evolution in their research.

With our new investigations on ocean fluxes and energy transport, as well as the numerical experiments about the ice shelf variability and its upstream coupling we tried to broaden the context of the manuscript and explain in more details which implications are likely and of interest for a broader environmentally concerned community.

- **Data & methodology:** This work uses a lot of very disparate datasets (ground-based, remote sensing and modeling). While I find the writing and organization hard to follow, the authors do include all the relevant data descriptions.

We tried to include all relevant data, which support the observations of the Midgardsormen migration. It is not a simple task to present all these different data in a concise form and still keep the manuscript short enough to be attractive to read. We tried to improve the writing in the Data section in order to allow easier reading.

- **Appropriate use of statistics and treatment of uncertainties:** Yes, the authors are careful about statistics.

- **Conclusions:** Overall, I found the conclusion that the ice shelf has thinned to be convincing and well documented. The inferences about atmospheric forcing from positive degree day estimates and a 20 year old plume model based on 4 CTD casts are not very convincing (or as well described).

There is no disadvantage in using old theories and models as long as they do not contradict basic physics. The advantage is that these approaches are simple and still provide solid answers to the questions we asked. The simple degree day model gives us the answer, that even for a large temperature increase (implying a considerably larger atmospheric energy transfer), the atmospheric conditions cannot explain the observed variability in ice thickness loss. The plume model provides information about the intensity of sub ice shelf fluxes, which is the basic parameter necessary for estimating oceanic energy exchange into the cavity. It is not the intension of this manuscript to explain detailed plume geometries, but providing basic potential causes for the observed thinning. Given that the data availability is very sparse for this region, the application of more sophisticated models without adequate input data would be even more questionable.

We changed several parts of the manuscript to improve the arguments about the oceanic forcing. Please refer to our comments regarding reviewer #2.

- **Clarity and context:** This paper is possibly Nature-worthy if it was easier to follow and written more concisely. The first time I read the paper I thought the shear-zone analysis was going to be the major point of the paper. But, the main conclusion is the long-term thinning (which is an interesting result), and the shear zone is just one tool used to derive thickness change.

We still regard the shear zone migration a major result of the paper, because without this mechanisms it would have been impossible to derive the long-term thinning and its temporal variability. However, we tried to improve the clarity of the manuscript by better balancing the text between data and findings.

I've included some specific comments for the first few pages. However, most of these are

editorial comments, so I did not continue to make the corrections for the latter part of the text. Throughout the text, the verb tenses are confusing, there are multiple typos and it doesn't seem to follow the Nature guidelines (for length, location of methodology, structure of abstract, or placement of figures).

Abstract: It is slightly confusing what the main question/problem is here. For example, "A considerable loss in ice thickness was observed" – is this your result or a previous observation? Per Nature guidelines, they like the summary paragraph to state the problem and then conclude with "Here we show..."

We changed the abstract in order to comply better to Nature guidelines and clarify the contribution this manuscript provides to the research of NE Greenland glacier conditions.

15: The transition from "Based on the migration of a surface feature" to "ice thickness and bedrock data" is awkward (and missing some verbs).

We changed this sentence to clarify our contribution.

17: "for producing" -> to produce

Changed

18: This statement seems overly confident...your results definitely suggest this conclusion, but not definitively.

We agree with the reviewer and have revised the abstract, using the formulation: "increased ocean heat flux is the most plausible cause of the observed thinning".

27: This sentence needs more detail – could contribute to 1.1 m of sea level rise (under what conditions? Over what time interval? Based on what?)

Here we only state the potential contribution if the ice sheet disappears according to Morlighem et al., 2014. This is a common information to relate ice volumes of parts of the ice sheet to maximum consequences. We now included "in the unlikely case of complete melt down".

28: Add some more detail to this implication – how much does it contribute to freshwater flux? A significant amount? It's a pretty slow moving glacier that does not calve icebergs frequently.

This sentence refers to the entire section which provides a considerable fresh water amount. Calving is not a valid measure in this context, because subglacial melt is the dominating effect for 79 North Glacier and an important contribution at Zachariae Isstrøm. More details are now provided in this context.

30: early?

Changed

31: Need references for this statement

Reference added

36. Missing a period

Added

37: “strong increase in ice flux” – by how much?

We added this information in the manuscript.

38: Glacier’s

Done

39-42: This paragraph doesn’t seem necessary, especially given the strict word limit

The warming trends are essential for the energy availability in the region. We therefore prefer to keep this information in the manuscript. However, we removed the last sentence to shorten the paragraph.

44: “extensive cavity beneath the ice shelf” – does this just mean that the shelf is floating, or that there is a big bed depression under the ice shelf?

"Extensive" in this context described a bed depression with large regions being rather deep. The formulation has been changed to "Seismic measurements have revealed a deep ocean cavity beneath the ice shelf"

48: represents “a” remarkable...

Included

53: The transition to this last sentence is awkward

This sentence was removed.

Fig 1: Suggest changing the color of one of the “light blue” features and adding the location of the grounding line.

We added the approximate grounding line position in the figure. We keep the light blue colours, because they indicate the region of the Midgardsormen feature and the relevant part of the profiles used.

60: as “a” red dot

Added

94: How were the ATM data smoothed? Why?

The ATM data have been resampled to the resolution of the airborne radar sampling of 34 m, by using the arithmetic mean of the samples. Thus, it was easier to compare.

106: “was tracked” and “have been selected” ?

Changed

Reviewers' comments:

Reviewer #1 (Remarks to the Author):

I have already (in the first review) stated that the paper is very well written and well organized. My main concern was a stronger support of the statement given in the manuscript and that the authors should include modeling. This is taken care of in the updated version, and have no further major issues that needs to be addressed before publication.

Reviewer #2 (Remarks to the Author):

The authors have dealt thoroughly with my comments (either by modifying the text, or presenting a convincing rebuttal), and I do not have any further corrections or additions.

Reviewer #3 (Remarks to the Author):

This revised manuscript is certainly much clearer to read and the results are presented in a convincing manner. While I trust the results, and agree that they are derived innovatively, I have misgivings about the appropriateness of some of the references, organization of the results section, and clarity of some of the text. Perhaps I just misread the intent in the abstract, but I do find that the statement "ice modeling shows a significant thinning upstream of the grounding line" (line 20-21) to be contradictory of the actual results "the increased fluxes will not reach far into the ice sheet during the next century" (line 338). My points can all be fixed with careful editing.

Introduction:

25: (75 North Glacier) has the largest....

29: "complete meltdown" does not sound very glaciological or scientific. Do you mean if all the ice in the catchment drained to the ice shelf? Or it all melted? I don't see how the reference (Morlighem et al) predicts this 1.1 m value.

32: I find that this sentence contradicts the reference that it cites. "...the mass balance of the NEGIS was long considered to be close to equilibrium" vs "Continued thinning at these rates would unground the ice plain within 40 years or less, possibly leading to an increase in glacier discharge, and additional thinning of the NEGIS" (Thomas et al., page 156).

40-41: This transition is awkward (and 41-43 seems out of place to me). You just finished writing about how the upstream ice flux caused Zacharaie's disintegration and go straight into how the oceans are warming (the implication being that they are melting 79N ice shelf). It would help if these different processes were spelled out more explicitly.

- "are likely" to affect the stability of 79N makes a lot of assumptions: that sea ice impacts calving rates, and that both air and ocean temps cause ice shelf break up, and that ice shelf breakup impacts glacier stability. I think the language needs to be softer here, or the links between these processes made clearer. "could affect the stability" would help.

51: Fig 6? List figures in chronological order.

52: "shown in Figure 1" and described in the Supplemental Material (?)

55: "used an ice sheet model"

In general, I think the Introduction still needs to be better organized. If your key points are that the ice shelf is thinning and this is causing upstream thinning, then introducing these connections

and processes here is important (impact of grounding line retreat on ice discharge, etc).

Results:

73: reference

72-81: This paragraph is fairly cumbersome to read as it is kind of a list of different thinning rates. Can't this be described in a table more clearly (Table S1 covers most of it).

82-90: This seems like methodology

93: Fig 4 (before Figure 2 or 3?)

94-115: This paragraph is well written and exactly what should be here. But, the preceding paragraph takes away from the excitement and innovation of this paragraph.

132: Co-registered Landsat scenes reveal...

141: Can you add another column to Table 1 that shows the long-term remote sensing thinning rates (merge a few rows together to show the integrated results)?

149: Supplementary Material (capitalize).

166: ice thickness reduction -> ice thinning

177-182: punctuation of the bulleted list (semicolons after each bullet and a period at the end)

329: The transition between grounding line retreat and ice thickness is confusing to me. Seems like the topic jumped quite a bit.

338: The conclusion reached here, that thinning does not reach far into the ice sheet seems somewhat contrary to what is described in the abstract (line 21-22): "ice modeling shows a significant thinning upstream of the grounding line in response".

Dear Reviewers,

Thank you very much for the time and energy invested in reviewing the manuscript.

We carefully revised the manuscript, evaluating all suggestions provided by Reviewer #3. The corrections are described point-by-point in response to the reviewer's comments.

Reviewers' comments:

Reviewer #1 (Remarks to the Author):

I have already (in the first review) stated that the paper is very well written and well organized. My main concern was a stronger support of the statement given in the manuscript and that the authors should include modeling. This is taken care of in the updated version, and have no further major issues that needs to be addressed before publication.

Reviewer #2 (Remarks to the Author):

The authors have dealt thoroughly with my comments (either by modifying the text, or presenting a convincing rebuttal), and I do not have any further corrections or additions.

Reviewer #3 (Remarks to the Author):

This revised manuscript is certainly much clearer to read and the results are presented in a convincing manner. While I trust the results, and agree that they are derived innovatively, I have misgivings about the appropriateness of some of the references, organization of the results section, and clarity of some of the text. Perhaps I just misread the intent in the abstract, but I do find that the statement "ice modeling shows a significant thinning upstream of the grounding line" (line 20-21) to be contradictory of the actual results "the increased fluxes will not reach far into the ice sheet during the next century" (line 338). My points can all be fixed with careful editing.

Introduction:

25: (75 North Glacier) has the largest....

changed

29: "complete meltdown" does not sound very glaciological or scientific. Do you mean if all the ice in the catchment drained to the ice shelf? Or it all melted? I don't see how the reference (Morlighem et al) predicts this 1.1 m value.

We changed it into "a complete loss of the ice sheet sector", indicating that the process does not need to be melt, but could also be by mass transport into the ocean. The reference to Morlighem et al. (2014) was a mix-up, it should be Mougnot et al. (2015). We corrected this.

32: I find that this sentence contradicts the reference that it cites. "...the mass balance of the NEGIS was long considered to be close to equilibrium" vs "Continued thinning at these

rates would unground the ice plain within 40 years or less, possibly leading to an increase in glacier discharge, and additional thinning of the NEGIS" (Thomas et al., page 156).

We considered only the grounded part of NEGIS. Now we describe this more precisely with a "close to equilibrium state" for the upstream part and observed thinning in the coastal regions. Thomas et al, page 155: "Surface lowering predominated between 1994 and 1999 "in the coastal region", at rates of about 0.1m a^{-1} , with almost exact balance further inland.", and page 156: "Slow, erratic thinning predominated over the ice plain in 1994 and 1999, with approximate balance over the inland 40km."

40-41: This transition is awkward (and 41-43 seems out of place to me). You just finished writing about how the upstream ice flux caused Zachariae's disintegration and go straight into how the oceans are warming (the implication being that they are melting 79N ice shelf). It would help if these different processes were spelled out more explicitly.

We now explained the potential causes for the observations at Zachariæ Isstrøm in more detail, in relation to the arguments of Khan et al., 2014. They discuss similar potential reasons for the break-up and the involved glacier speed-up as we do in lines 41-43. Therefore, we now point out these processes and then raise the question about their impact for the northern neighbour, 79 North Glacier.

- "are likely" to affect the stability of 79N makes a lot of assumptions: that sea ice impacts calving rates, and that both air and ocean temps cause ice shelf break up, and that ice shelf breakup impacts glacier stability. I think the language needs to be softer here, or the links between these processes made clearer. "could affect the stability" would help.

We "softened" the statement and brought it into a better context.

51: Fig 6? List figures in chronological order.

This is due to the structure of the journal, presenting the Data and Methods after the Results and Discussion. We do not want to break-up Fig. 6, presenting the photograph in the Introduction and the location of the measurements in the Methods chapter and thus creating an extra figure. Because the reference is to a photograph and not to a result, this should be acceptable. But we leave this decision to the editor.

52: "shown in Figure 1" and described in the Supplemental Material (?)

We changed the sentence and now also mention the Supplemental Material here.

55: "used an ice sheet model"

We now write "an ice dynamic model", as it is a coupled ice sheet – ice shelf model.

In general, I think the Introduction still needs to be better organized. If your key points are that the ice shelf is thinning and this is causing upstream thinning, then introducing these connections and processes here is important (impact of grounding line retreat on ice discharge, etc).

According to other comments of the reviewer we changed this paragraph of the introduction. Now the processes are more clearly explained and better related to the case of 79 North Glacier.

Results:

73: reference

added

72-81: This paragraph is fairly cumbersome to read as it is kind of a list of different thinning rates. Can't this be described in a table more clearly (Table S1 covers most of it).

We now shifted the numerical values in a new table in the Supplementary Material and focus only on the main finding of a change in ice thickness of 98.1 m between 1997 and 2016 and a respective change rate of 5.16 m/yr.

82-90: This seems like methodology

We moved this part to the methods chapter and some information into the Supplementary Material, describing the error calculation of the ice thickness changes.

93: Fig 4 (before Figure 2 or 3?)

The order is now correct due to the changed structure of the results chapter.

94-115: This paragraph is well written and exactly what should be here. But, the preceding paragraph takes away from the excitement and innovation of this paragraph.

We now changed the order of the presentation of the results, focussing first on the new findings derived from the grounding line migration. The comparison with the regional and long-term changes is presented after this section.

132: Co-registered Landsat scenes reveal...

changed

141: Can you add another column to Table 1 that shows the long-term remote sensing thinning rates (merge a few rows together to show the integrated results)?

The long-term thinning rates are rather uniform and the mean value is discussed in the text. In addition, we now present these values in an additional table in the Supplementary Material. Therefore an additional presentation in Tab. 1 is not required in our opinion.

149: Supplementary Material (capitalize).

changed

166: ice thickness reduction -> ice thinning

changed

177-182: punctuation of the bulleted list (semicolons after each bullet and a period at the end)

done

329: The transition between grounding line retreat and ice thickness is confusing to me. Seems like the topic jumped quite a bit.

We now connected the numerical results of grounding line retreat and ice sheet thinning to the question about the probability of an ice shelf disintegration. Thus, we can demonstrate the linkage, how the observed thickness change indicates potential future consequences on the entire glacier system.

338: The conclusion reached here, that thinning does not reach far into the ice sheet seems somewhat contrary to what is described in the abstract (line 21-22): “ice modeling shows a significant thinning upstream of the grounding line in response”.

It was difficult to present all important details in the length-restricted abstract. We now shortened some parts of the abstract to include the local character of the ice sheet thinning. The thinning found in the model results is significant at a local scale. However, for the short period of model simulations it does not extend too far inland. For longer simulation periods the model and the boundary conditions needs to be more sophisticated, which is out of the scope of this manuscript.